# Breaking BatchNorm Barriers for Noise-driven Data Free Knowledge Distillation

## Abstract

Distillation using Gaussian noise is the simplest instantiation of data-free knowledge distillation: it uses no auxiliary generator, no synthesized images, and no proxy data. The idea is to sample inputs from a standard Gaussian and match teacher-student outputs. In practice, however, this approach is fragile: its behavior varies sharply across architectures and is tightly coupled to the teacher's normalization choices. In this work, we systematically study when and why Gaussian-noise distillation succeeds or fails across model families, normalization schemes, and scales from CIFAR-10 to ImageNet-100, and we identify the main factors that control its stability and effectiveness. Building on these insights, we propose NormShift-KD, a normalization-aware framework for noise-driven distillation with two instantiations tailored to the teacher's normalization: (i) for BatchNorm teachers, we pair current-statistics (CS) inference with rejection sampling to correct the class-imbalance that BN teachers exhibit on Gaussian inputs; (ii) for LayerNorm and GroupNorm teachers, we introduce a lightweight batch-alignment wrapper that restores the inter-sample coupling these per-sample normalizers lack, enabling noise-driven distillation from non-BatchNorm teachers for the first time. We further conduct a targeted BatchNorm ablation, progressively replacing BatchNorm in the teacher to map how transfer quality degrades and which components matter most, and we analyze how the student's architecture and normalization interact with the teacher (CNN/BatchNorm vs. Transformer/LayerNorm). Finally, we provide a theoretical explanation for the failure modes observed in ViT-style models under Gaussian-noise inputs, making noise-driven distillation more interpretable and more broadly usable.

## 1 Introduction

Knowledge distillation compresses a high-capacity teacher into a smaller student by training the student to match the teacher's predictions (Hinton et al., 2015). In the data-free setting, the teacher's training data is unavailable due to privacy, licensing, or governance constraints, and distillation must proceed using only the teacher model itself (Lopes et al., 2017). Among data-free approaches, distillation from synthetic inputs sampled from a simple distribution (e.g., i.i.d. Gaussian noise) is arguably the most minimal: it removes the need for auxiliary generators (Chen et al., 2019; Micaelli & Storkey, 2019; Fang et al., 2021) or access to any real/proxy samples (Yin et al., 2020). Despite this appeal, Gaussian-noise-driven distillation is widely observed to be unreliable, with outcomes that vary sharply across architectures, training configurations, and normalization choices (Raikwar & Mishra, 2022).

Prior work has established two key failure modes of noise-driven distillation. First, Raikwar & Mishra (Raikwar & Mishra, 2022) demonstrated that teachers with Batch Normalization (BN) (Ioffe & Szegedy, 2015) can provide useful supervision under Gaussian noise *only* when the teacher is run in CS mode (using current statistics), but fail when run in RS mode (using running statistics). This occurs because the distribution of activations under noise deviates substantially from the running statistics computed during original training, causing a covariate shift that produces uninformative soft targets. Second, the data-free knowledge distillation literature has predominantly focused on BN-based architectures (Yin et al., 2020; Fang et al., 2021; Choi et al., 2020), with limited exploration of teachers using per-sample normalization schemes such as LayerNorm (LN) (Ba et al., 2016) or GroupNorm (GN) (Wu & He, 2018). These normalization layers, which are standard

in Vision Transformers (Dosovitskiy et al., 2021) and many modern architectures, lack the batch-level coupling that enables noise-driven distillation with BN teachers. In this paper, we first provide a comprehensive analysis of why RS-mode teachers produce weak supervision under noise: predictions become highly imbalanced across classes and exhibit low diversity, and internal activations deviate substantially from those induced by real data. In contrast, CS-mode BN introduces a form of *batch coupling*—samples in a minibatch are normalized using shared per-channel statistics—which preserves inter-sample structure under noise and yields more coherent soft targets for the student.

Motivated by these observations, we propose **NormShift-KD**, a normalization-aware framework for noise-driven distillation that targets the distinct failure modes induced by different teacher normalizations. For BatchNorm teachers, we combine current-statistics (CS) inference with *rejection sampling* (RejS), a class-balancing procedure that subsamples noise batches toward a uniform pseudo-label distribution before distillation. For LayerNorm and GroupNorm teachers, we wrap each per-sample normalization layer with a lightweight *batch-alignment* step that aligns normalized activations using batch-level per-channel statistics, together with a *soft alignment* variant that interpolates between pure per-sample normalization and full batch alignment via a single mixing coefficient. RejS is applied in both instantiations; CS-mode inference and batch alignment are the normalization-specific components that differ across teachers. The two instantiations share a common diagnosis—the teacher's behavior under Gaussian noise is governed by its normalization—and together extend noise-driven distillation to both BatchNorm and non-BatchNorm teachers without introducing generators or real data.

Beyond proposing a practical fix, we provide a detailed experimental analysis of the factors that govern success in noise-driven distillation. We conduct a targeted BN ablation by progressively replacing BN layers with LN in a controlled ResNet family, revealing how transfer degrades as batch coupling is removed and which parts of the network matter most. We also analyze the role of student architecture and normalization, including Transformer-based students, and we provide a theoretical explanation for a key failure mode in ViT-style models (Dosovitskiy et al., 2021): under random noise, self-attention becomes high-entropy and near-uniform, effectively reducing token interactions to averaging and producing weak class-discriminative supervision (Touvron et al., 2021; Habib et al., 2023). Finally, we demonstrate that the resulting insights and methods scale from CIFAR (Krizhevsky & Hinton, 2009) to ImageNet (Deng et al., 2009).

The key contributions are listed below:

- We identify and quantify why RS-mode teachers and LN/GN-based teachers fail to provide useful supervision under Gaussian-noise distillation, using both activation-space and prediction-space diagnostics.

- We perform a controlled BatchNorm ablation by progressively replacing BatchNorm layers with LayerNorm, mapping how transfer quality degrades as batch coupling is removed.

- We propose **NormShift-KD**, a normalization-aware framework with two instantiations: a rejection-sampled CS-mode procedure for BatchNorm teachers, and a batch-alignment wrapper (with a soft variant) for LayerNorm and GroupNorm teachers. The framework is fully data-free and generator-free, and to our knowledge yields the first noise-driven distillation results from non-BatchNorm teachers.

- We analyze student normalization and architecture effects, and provide a theoretical account of why Gaussian-noise supervision is weak for ViT-style attention mechanisms.

## 2 Background and Problem Setup

### 2.1 Related Work

**Knowledge distillation.** Knowledge distillation (KD) (Hinton et al., 2015) trains a compact student network $S$ to mimic a larger teacher network $T$ by matching their output distributions on a transfer set. The standard objective combines a soft distillation loss, which aligns temperature-scaled softmax outputs, with an optional hard-label cross-entropy term. KD has become a foundational technique for model compression (Gou

et al., 2021). Recent work has refined the classical formulation: Decoupled Knowledge Distillation (Zhao et al., 2022) separates target-class and non-target-class contributions to improve knowledge transfer, while DIST (Huang et al., 2022) introduces correlation-based losses to handle stronger teachers. For Vision Transformers, DeiT (Touvron et al., 2021) demonstrated effective cross-architecture distillation from CNN teachers using a dedicated distillation token. More recently, VL2Lite (Jang et al., 2025) extends KD to leverage vision-language models for enhancing lightweight networks, and Frank & Davis (Frank & Davis, 2025) systematically study what properties make datasets effective for knowledge transfer.

**Data-free knowledge distillation.** When the teacher's training data is unavailable due to privacy, licensing, or storage constraints, data-free KD methods synthesize a surrogate transfer set. Existing approaches fall into two categories: (i) *generator-based* methods train an auxiliary network to produce synthetic images that match batch normalization statistics (Chen et al., 2019; Micaelli & Storkey, 2019; Fang et al., 2021) or maximize class diversity (Yin et al., 2020; Choi et al., 2020); (ii) *noise-based* methods directly use random samples (e.g., Gaussian noise) without any learned generator (Nayak et al., 2019; Raikwar & Mishra, 2022). While generator-based approaches dominate current benchmarks, noise-based methods are appealing for their simplicity and zero auxiliary cost.

**Normalization layers.** Batch Normalization (BN) (Ioffe & Szegedy, 2015) normalizes activations using statistics computed across the minibatch, introducing implicit coupling between samples. During training, BN computes per-channel mean and variance from the current batch; during inference (evaluation mode), it uses exponential moving averages accumulated over training, stored as *running mean* and *running variance* parameters. In contrast, Layer Normalization (LN) (Ba et al., 2016) and Group Normalization (GN) (Wu & He, 2018) compute statistics per-sample, removing any batch-level dependence. LN and GN are standard in Transformers (Vaswani et al., 2017; Dosovitskiy et al., 2021) and scenarios with small or variable batch sizes.

**Positioning.** Data-free distillation comprises two distinct research questions. The first asks how to *synthesize* a transfer set that approximates the teacher's training distribution: generator-based methods Chen et al. (2019); Micaelli & Storkey (2019); Fang et al. (2021) and model-inversion methods Yin et al. (2020) optimize inputs or train auxiliary networks against teacher statistics, and progress in this line is measured by the fidelity of the synthesized set. The second asks what the teacher itself reveals when queried with unstructured inputs: noise-driven methods Nayak et al. (2019); Raikwar & Mishra (2022) treat the teacher as the sole object of study and ask which of its properties make distillation feasible without any input synthesis. The two questions involve different mechanisms, different cost models, and different sources of difficulty—synthesis methods are bottlenecked by generator quality, noise-driven methods by the teacher's behavior on out-of-distribution inputs—and progress on one does not transfer to the other. Our work is in the second line. Within it, prior analysis is limited to BatchNorm teachers in CS mode Raikwar & Mishra (2022); the behavior of LayerNorm and GroupNorm teachers, the role of student normalization, and the failure modes of attention-based teachers under noise have not been characterized. We address these gaps and introduce **NormShift-KD**, a normalization-aware framework that strengthens noise-driven distillation in both regimes: it pairs CS-mode inference with rejection sampling to mitigate the class-imbalance of BatchNorm teachers under Gaussian inputs, and it introduces a batch-alignment wrapper that, to our knowledge, yields the first noise-driven distillation method that succeeds with non-BatchNorm teachers. Our baselines are therefore the noise-driven and data-dependent Gaussian methods that share this problem formulation Raikwar & Mishra (2022); Frank & Davis (2025).

## 2.2 Problem Setup

We study noise-driven data-free knowledge distillation for image classification. Let $T$ denote a pretrained teacher and $S$ a randomly initialized student, with no access to the teacher's training data. Synthetic inputs are sampled as $x \sim \mathcal{N}(0, I)$, and the student is optimized to match the teacher's predictions.

**Distillation objective.** Following Raikwar & Mishra (2022), we train the student using only the teacher's soft predictions. When the teacher contains BatchNorm layers in CS mode, its outputs depend not only on the input $x$ but also on the batch context $\mathcal{B} = \{x_1, \ldots, x_m\}$. We denote teacher logits as $z_T(x; \mathcal{B})$ to make

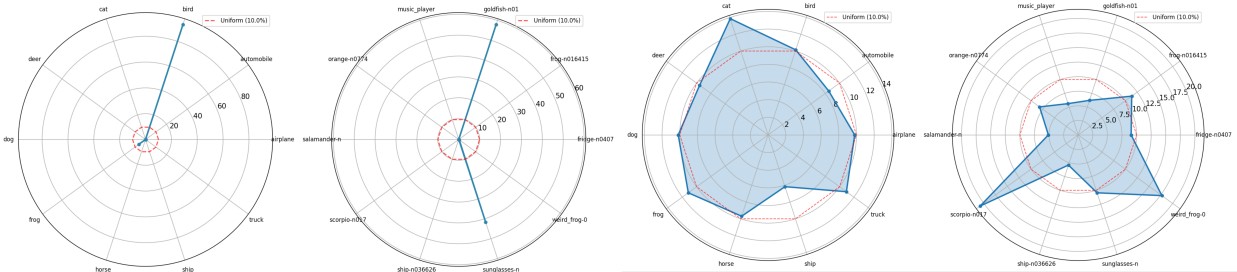

Figure 1: **Teacher mode controls prediction diversity under Gaussian noise.** Class-wise prediction distributions of a ResNet-34 teacher on Gaussian noise, shown for RS mode (first two plots) and CS mode (last two plots). From left to right: CIFAR-10 and ImageNet-10. In RS mode, BatchNorm uses running statistics estimated from real images, leading to highly imbalanced, low-entropy predictions (degenerate supervision). In CS mode, BatchNorm uses minibatch statistics computed on noise, restoring class coverage and producing substantially more informative soft targets for noise-driven distillation. Experimental details are provided in Section 3.1 and Appendix F.

this dependence explicit. The objective minimizes the expected KL divergence:

$$\min_S \; \mathbb{E}_{\mathcal{B} \sim \mathcal{D}_{\text{noise}}^m} \left[ \frac{1}{m} \sum_{x \in \mathcal{B}} \text{KL}\big(p_T(\cdot \mid x, \mathcal{B}) \,\|\, p_S(\cdot \mid x)\big) \right], \tag{1}$$

where $\mathcal{D}_{\text{noise}} = \mathcal{N}(0, I)$, each minibatch $\mathcal{B} = \{x_i\}_{i=1}^m$ consists of $m$ i.i.d. samples, $p_T(y \mid x, \mathcal{B}) = \sigma(z_T(x; \mathcal{B})/\tau)$ and $p_S(y \mid x) = \sigma(z_S(x)/\tau)$ are the teacher and student softmax distributions, and $\tau > 0$ is a temperature parameter. We write the student's distribution as $p_S(\cdot \mid x)$; when the student contains BatchNorm layers in training mode—as in all our main experiments (Section 5.2)—its logits additionally depend on the batch context, $z_S(x; \mathcal{B})$, and Eq. (1) is understood with $p_S(\cdot \mid x, \mathcal{B})$. For students with per-sample normalization (LayerNorm, GroupNorm), outputs are batch-invariant and the conditioning is omitted; the consequences of batch invariance are analyzed in Section 5.2.

**Teacher mode.** The teacher parameters remain frozen throughout distillation. We run the teacher in *CS mode*, where BatchNorm layers normalize activations using statistics from the current minibatch rather than stored running statistics (RS mode). This choice is critical: current batch statistics yield informative supervision under Gaussian noise, whereas RS mode produces degenerate predictions (Raikwar & Mishra, 2022). We analyze this failure mode in upcoming sections.

Throughout the paper, $\mathbb{E}[\cdot]$ denotes expectation and $\mathbb{V}[\cdot]$ denotes variance; subscripts indicate the indices over which these are computed.

## 3 Why Gaussian-Noise KD Depends on Normalization

### 3.1 The Role of Batch Statistics in Noise-Driven KD

A common default in distillation is to run the teacher in evaluation mode. Under Gaussian-noise inputs, this choice severely degrades supervision quality, even when the teacher is highly accurate on real data.

When a BatchNorm-based teacher processes Gaussian noise in RS mode, predictions become highly imbalanced: a small subset of classes dominates while many receive near-zero probability mass. This effect persists across datasets with varying class counts, as shown for CIFAR-10 and ImageNet-10 in Fig. 1 (ImageNet-100 in Appendix G). Predictive entropy drops sharply and the training signal collapses.

In evaluation mode, BN normalizes activations using running mean and variance accumulated from real training images. Gaussian noise induces a distribution shift in intermediate activations, causing normalization with mismatched statistics. Figure 4 (Appendix G) illustrates this mismatch in an early Conv–BN–ReLU

block: noise-induced activations concentrate in a narrow range that is poorly aligned with the mean/variance statistics learned from real data.

Formally, RS-mode BN uses fixed statistics $(\mu_0, \sigma_0^2)$ estimated during training. For pre-activation $u = w^\top x$, the normalized output is $\tilde{u} = (u - \mu_0)/\sqrt{\sigma_0^2 + \varepsilon}$. When the input distribution has $\mathbb{E}[u] = \mu_* \neq \mu_0$ or $\mathrm{Var}(u) = \sigma_*^2 \neq \sigma_0^2$, the normalized activations remain shifted and mis-scaled: $\mathbb{E}[\tilde{u}] = (\mu_* - \mu_0)/\sqrt{\sigma_0^2 + \varepsilon}$ and $\mathrm{Var}(\tilde{u}) = \sigma_*^2/(\sigma_0^2 + \varepsilon)$.

Running the teacher in CS mode replaces stored statistics with minibatch statistics computed on-the-fly. Given a minibatch $\{x_i\}_{i=1}^B$ with pre-activations $u_i = w^\top x_i$, CS-mode BN computes $\mu_B = \frac{1}{B}\sum_i u_i$ and $\sigma_B^2 = \frac{1}{B}\sum_i (u_i - \mu_B)^2$, yielding $\hat{u}_i = (u_i - \mu_B)/\sqrt{\sigma_B^2 + \varepsilon}$. As $B \to \infty$, $\mathbb{E}[\hat{u}] \to 0$ and $\mathrm{Var}(\hat{u}) \to 1$ regardless of the input distribution, re-centering activations to a canonical regime. This restores prediction diversity and yields informative soft targets (Raikwar & Mishra, 2022).

We investigate the distribution of teacher predictions when the input is Gaussian noise. Figure 1 visualizes this distribution for teachers trained on ImageNet-10 via radar plots. (ImageNet 100 radar plots are in Appendix G)

To quantify this effect at scale, we analyze a ResNet-34 trained on full ImageNet-1K in Table 1. Even at 1.2M samples—comparable to the original training set size—the prediction distribution remains highly non-uniform: the most frequent class receives 12,078 samples while the least frequent receives only 1 (ratio $> 12000\times$). Notably, the Max/Expected ratio stabilizes around $10\times$ regardless of sample count, indicating that this imbalance is a structural property of the teacher's mapping from Gaussian space to logits rather than a finite-sample artifact.

Table 1: Teacher prediction statistics over Gaussian noise (ResNet-34, ImageNet-1K, CS mode).

| Samples | Zero Classes | Min | Max |
|---|---|---|---|
| 4,000 | 241 (24.1%) | 0 | 41 |
| 10,000 | 125 (12.5%) | 0 | 103 |
| 100,000 | 13 (1.3%) | 0 | 985 |
| 1,200,000 | 0 (0%) | 1 | 12,078 |

The degradation at larger scales can be attributed to an inherent teacher bias: when processing Gaussian noise, the teacher exhibits a strong preference toward a subset of classes while largely ignoring others. This bias emerges from the learned weight structure and BatchNorm statistics, which together define decision boundaries that partition the Gaussian input space unevenly across classes. As the number of classes grows, this partitioning becomes increasingly imbalanced—certain classes occupy large volumes of the input space while others are effectively unreachable via standard Gaussian sampling.

The persistent $\sim 10\times$ concentration ratio at ImageNet-1K (Table 1) quantifies this bias: regardless of how many samples we draw, the teacher consistently favors the same classes by roughly an order of magnitude over uniform expectation. At 1.2M samples, the most favored class receives 12,078 predictions while the least favored receives only 1, yielding a ratio exceeding 12,000$\times$. This structural bias cannot be mitigated by simply increasing sample count; it reflects how the teacher's learned representations carve up the high-dimensional noise manifold. Consequently, achieving balanced class coverage through rejection sampling becomes computationally prohibitive at ImageNet-1K scale, requiring massive oversampling factors that grow rapidly with the class imbalance, especially as the minimum class probability becomes very small.

### 3.2 Normalization Determines Batch Consistency Under Noise

**Definition 3.1** (Batch coupling). Let $f(x; \mathcal{B})$ denote a network's output for input $x$ when processed jointly with batch context $\mathcal{B} \ni x$. We call $f$ *batch-coupled* if there exist an input $x$ and contexts $\mathcal{B}_1, \mathcal{B}_2$ with $f(x; \mathcal{B}_1) \neq f(x; \mathcal{B}_2)$, and *batch-invariant* if $f(x; \mathcal{B}) = f(x)$ for all $\mathcal{B}$. CS-mode BatchNorm is batch-coupled, since all samples are normalized by shared minibatch statistics $(\mu_\mathcal{B}, \sigma_\mathcal{B}^2)$; LayerNorm and GroupNorm are

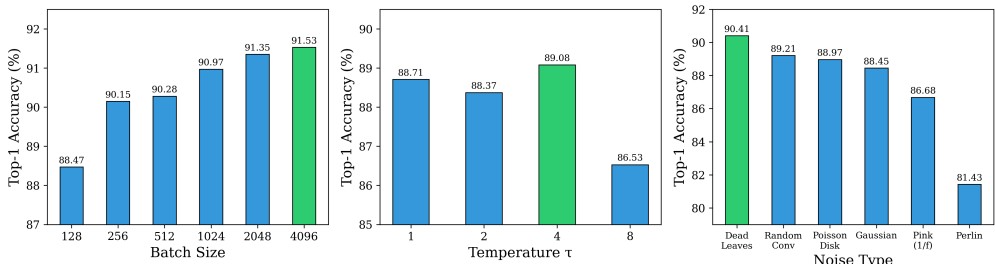

Figure 2: **Sensitivity of noise-driven distillation.** ResNet-34 → ResNet-18 on CIFAR-10. Best top-1 accuracy (%).

batch-invariant. Empirically, we measure coupling by the argmax flip rate and per-class probability variance of $f(x; \cdot)$ across resampled batch contexts (protocol in Appendix D, Table 12).

**Mechanism.** Shared batch statistics play two roles under Gaussian inputs. First, *re-centering*: normalizing with statistics computed on the current inputs returns activations to the regime in which downstream weights were trained ($\mathbb{E}[\hat{u}] \to 0$, $\mathrm{Var}(\hat{u}) \to 1$; Appendix G, Figure 4), preventing the covariate-shift collapse observed in RS mode. Second, a *shared reference frame*: because all samples in $\mathcal{B}$ are normalized identically, inter-sample differences are preserved through the teacher's discriminative features, spreading pseudo-labels across classes rather than concentrating them on a few; the $I(X; C)$ decomposition in Appendix C quantifies this effect. Coupling is not *sufficient*, however: Section 7.1 shows that ViT teachers fail under noise even when coupling is restored, owing to attention collapse.

BatchNorm in CS mode computes per-channel statistics across the minibatch and applies identical normalization to all samples, providing a shared reference frame even under random inputs. In contrast, LayerNorm and GroupNorm compute statistics independently per sample, eliminating batch coupling. Under Gaussian noise, this per-sample normalization reduces inter-sample consistency and yields unstable supervision. This distinction foreshadows a key failure mode: when BN is removed, noise-driven distillation fails unless batch coupling is reintroduced.

Noise-driven distillation as shown in Figure 2 is also sensitive to batch size—accuracy improves from 88.5% to 91.5% as batch size increases from 128 to 4096 (see Appendix A)—further evidence that stable batch statistics are critical for effective knowledge transfer.

## 4 The BatchNorm Bottleneck: A Systematic Study

Section 3.2 argued that batch coupling is essential for noise-driven distillation and anticipated that removing BatchNorm should cause failure. We now test this directly by progressively replacing BatchNorm layers in a ResNet-34 teacher with LayerNorm or GroupNorm, retraining the teacher on CIFAR-10, and evaluating distillation to a ResNet-18 student. Unlike BatchNorm, LayerNorm and GroupNorm compute statistics per sample, eliminating batch coupling entirely.

Table 2 presents the results. Two findings emerge. First, teacher accuracy degrades gradually as BatchNorm is replaced: pure LayerNorm teachers achieve 91.40% versus 95.72% for pure BatchNorm, reflecting the known advantage of BatchNorm for CNNs. Second, student accuracy under noise distillation degrades dramatically and non-monotonically. Pure LayerNorm teachers yield catastrophic failure (18.00%), despite being only 4% weaker as classifiers. Progressively reintroducing BatchNorm recovers performance, with the best result (88.72%) at 25% LayerNorm. GroupNorm exhibits a similar pattern.

These results confirm that batch coupling—not teacher accuracy—is the primary determinant of distillation success under noise. Coupling is necessary but not sufficient: Section 7.1 shows that attention-based teachers fail under noise even when coupling is restored, an orthogonal failure mode that batch alignment does not

address. This motivates a key question: can we restore batch coupling to LayerNorm(LN)/GroupNorm(GN) teachers without modifying their architecture or retraining? We address this in the next section.

## 5 NormShift-KD: A Normalization-Aware Framework

Sections 3 and 4 identified two failure modes in noise-driven distillation. BatchNorm teachers in CS mode produce informative activations under Gaussian inputs, but their predictions remain strongly class-imbalanced, concentrating mass on a small subset of classes. LayerNorm and GroupNorm teachers fail more severely: per-sample normalization eliminates batch coupling entirely, and pure LN teachers yield only 18.00% student accuracy versus 88.08% for BN teachers, despite comparable real-data accuracy. We introduce *NormShift-KD*, a normalization-aware framework that addresses both failure modes through two components. A shared component, *rejection sampling* (RejS), corrects the class-imbalance that arises across both teacher regimes. A normalization-specific component then adapts to the teacher: CS-mode inference for BatchNorm teachers, or a batch-alignment wrapper for LayerNorm and GroupNorm teachers. Neither component requires architectural changes, retraining, or access to real data.

**Rejection sampling (RejS).** Both instantiations share a class-balancing step applied to every noise batch before distillation. Given a target batch size $B$ and an oversample factor $k$, we draw $kB$ inputs $x \sim \mathcal{N}(0, I)$, query the frozen teacher for pseudo-labels, and subsample $B$ inputs whose pseudo-label histogram best approximates the uniform distribution over classes. The procedure introduces a single hyperparameter, the oversample factor $k$, which increases teacher-inference cost by a factor of $k$ (forward passes only; no gradients are required); we use $k = 4$ throughout. RejS addresses the class imbalance documented in Figure 1 and Table 1. Appendix C.1 isolates its contribution under a matched protocol; the remaining improvement of NormShift-KD–BN over the Gaussian+CS baseline (Raikwar & Mishra, 2022) is attributable to the larger batch size. The required $k$ grows with the class imbalance, rendering rejection sampling computationally prohibitive at ImageNet-1K scale (Section 3.1, Table 1).

**CS-mode inference.** For completeness, we restate the BatchNorm-specific component, introduced by Raikwar & Mishra (2022) and analyzed in Section 3.1: during distillation, the frozen teacher's BatchNorm layers normalize with statistics computed on the current noise minibatch, $(\mu_{\mathcal{B}}, \sigma_{\mathcal{B}}^2)$, rather than with stored running statistics. NormShift-KD–BN combines CS-mode inference with RejS; NormShift-KD–LN replaces CS mode with the batch-alignment wrapper of Section 5.1 and retains RejS.

### 5.1 Batch-Aligned LayerNorm: The LN/GN Component

Table 2: Effect of progressively replacing BatchNorm with LayerNorm (LN) or GroupNorm (GN) in a ResNet-34 teacher on CIFAR-10. We replace BN layers sequentially from later (near output) to earlier (near input) stages; percentages indicate the fraction of BN layers converted. Student is ResNet-18 trained via Gaussian-noise distillation in CS mode.

| | % of BN layers replaced | | | | |
|---|---|---|---|---|---|
| | 0%
(pure BN) | 25% | 50% | 75% | 100%
(pure LN/GN) |
| *Teacher accuracy on real data (%)* | | | | | |
| BN → LN | 95.72 | 95.38 | 95.26 | 94.11 | 91.40 |
| BN → GN | 95.72 | 95.29 | 95.55 | 94.78 | 93.82 |
| *Student accuracy after noise distillation (%)* | | | | | |
| BN → LN | 88.08 | **88.72** | 83.38 | 77.51 | 18.00 |
| BN → GN | 88.08 | 87.45 | 79.81 | 68.21 | 11.90 |

Standard LayerNorm computes statistics over channels and spatial dimensions for each sample independently:

$$\text{LN}(x_b) = \gamma \odot \frac{x_b - \mu_b}{\sqrt{\sigma_b^2 + \epsilon}} + \beta, \tag{2}$$

where $(\mu_b, \sigma_b^2)$ depend only on sample $b$ and $(\gamma, \beta)$ are learned affine parameters. This batch-invariance—where the teacher's output for any input is independent of other samples in the minibatch—is precisely what causes distillation failure under noise.

Batch-aligned LayerNorm reintroduces inter-sample coupling by inserting a batch-level normalization step between per-sample normalization and the affine transformation. Given a minibatch $x \in \mathbb{R}^{B \times C \times H \times W}$, we first compute the non-affine per-sample output $\tilde{x} = \text{LN}(x; \text{affine} = \text{false})$. We then compute per-channel statistics across all samples and spatial locations:

$$\bar{\mu}_c = \mathbb{E}_{b,h,w}[\tilde{x}_{b,c,h,w}], \qquad \bar{\sigma}_c^2 = \mathbb{V}_{b,h,w}[\tilde{x}_{b,c,h,w}], \tag{3}$$

and apply batch-level normalization followed by the original affine parameters:

$$y = \gamma \odot \frac{\tilde{x} - \bar{\mu}}{\sqrt{\bar{\sigma}^2 + \epsilon}} + \beta. \tag{4}$$

This construction ensures that for each channel $c$, the batch-aligned activations satisfy $\mathbb{E}_{b,h,w}[\hat{x}] = 0$ and $\mathbb{V}_{b,h,w}[\hat{x}] \approx 1$ with statistics shared across all samples in the minibatch—directly mirroring BatchNorm's CS-mode behavior. The two-stage design is deliberate: operating on already-normalized per-sample outputs isolates the batch coupling effect from sample-specific scale and shift, preserving the teacher's learned representations while enabling cross-sample coordination.

**Soft batch alignment.** Full batch alignment may be unnecessarily aggressive in some settings. We introduce a soft variant that interpolates between per-sample and batch-aligned normalization:

$$\hat{x}_{\text{soft}} = (1 - \alpha) \cdot \tilde{x} + \alpha \cdot \hat{x}, \qquad y = \gamma \odot \hat{x}_{\text{soft}} + \beta, \tag{5}$$

where $\alpha \in [0, 1]$ controls the strength of batch coupling. Setting $\alpha = 0$ recovers standard LayerNorm, while $\alpha = 1$ yields full batch alignment.

We wrap every LayerNorm (or GroupNorm) layer in the frozen pretrained teacher. The wrapper introduces no trainable parameters and requires only two additional reductions (mean, variance) per layer per forward pass—negligible computational overhead. Importantly, batch-aligned LayerNorm makes the teacher's outputs batch-conditional: the prediction for input $x$ now depends on other samples through the shared statistics $(\bar{\mu}_c, \bar{\sigma}_c^2)$. By the analysis in subsection 5.2, batch-invariant students face irreducible approximation error when learning from batch-conditional teachers (Proposition 5.1). Section 5.2 details the resulting requirement on the student architecture.

## 5.2 Student Architecture Requirements

For LayerNorm and GroupNorm teachers, NormShift-KD's batch-alignment component makes the teacher's outputs batch-conditional. A natural question is whether the student architecture matters. We show that batch-invariant students (using LayerNorm or GroupNorm) face a fundamental limitation.

**Proposition 5.1** (Irreducible Error for Batch-Invariant Students). *If $p_T(\cdot \mid x, \mathcal{B}_1) \neq p_T(\cdot \mid x, \mathcal{B}_2)$ for some input $x$ and batch contexts $\mathcal{B}_1, \mathcal{B}_2$, then no batch-invariant student can achieve zero expected KL divergence. The minimum achievable loss equals $I(Y; \mathcal{B} \mid X = x)$—the mutual information between predictions and batch context.*

This mutual information is strictly positive when BatchNorm injects batch-dependent variation, establishing an irreducible error floor regardless of student capacity. We therefore use BatchNorm-based students in all NormShift-KD experiments.

Empirically, we validate this by applying batch alignment to ConvNeXt-LN students: accuracy recovers from 10% to 82.2% (Table 4). ViT-Tiny with batch alignment achieves only 14%, revealing additional challenges from attention collapse (Section 7.1). Full experimental details in Appendix D.

Table 3: Noise-driven data-free distillation accuracy (%) on CIFAR-10. DDG: data-dependent Gaussian.

| Teacher
*Teacher Acc.* | BN Teacher | | | | LN Teacher | | | |
| | R34
*95.79* | | WRN28-10
*95.12* | | R34
*91.40* | | WRN28-10
*94.54* | |
| Method | R18 | MV2 | WRN16-8 | R18 | R18 | MV2 | WRN16-8 | R18 |
| Gaussian + RS | 13.29 | 12.53 | 18.39 | 16.22 | 18.10 | 12.78 | 10.38 | 10.42 |
| DDG Frank & Davis (2025) | 65.42 | 62.75 | 68.19 | 66.72 | 18.72 | 13.82 | 11.21 | 10.98 |
| Gaussian + CS Raikwar & Mishra (2022) | 86.24 | 82.79 | 79.82 | 85.31 | 18.00 | 12.82 | 10.32 | 10.48 |
| **NormShift-KD (ours)** | **92.44** | **89.77** | **87.58** | **89.17** | **79.56** | **75.72** | **74.42** | **79.37** |

Table 4: Test accuracy on CIFAR-10 for LayerNorm students distilled from a ResNet-34 (BatchNorm) teacher using Gaussian noise. Batch Alignment recovers learning for ConvNeXt but not for ViT, revealing architecture-specific challenges beyond batch coupling.

| Student | Norm | Batch
Align. | Test
Acc. (%) |
|---|---|---|---|
| CNXt-Small | LN | ✗ | 10.0 |
| CNXt-Small | LN | ✓ | **82.2** |
| ViT-Tiny | LN | ✗ | 10.0 |
| ViT-Tiny | LN | ✓ | 14.0 |
| ViT-Tiny
(cached targets) | LN | ✓ | 17.0 |

**Student evaluation protocol.** During distillation the student's BatchNorm layers operate in training mode, so the running statistics they accumulate reflect Gaussian-noise activations rather than real data. We therefore report all student accuracies with BatchNorm in training mode on real test minibatches, i.e., using current test-batch statistics. This is consistent with the evaluation practice in noise-driven distillation: Raikwar & Mishra (2022) recalibrate BatchNorm running statistics by forwarding the (unlabeled) test set through the model before measuring accuracy in evaluation mode. Both protocols replace noise-accumulated running statistics with statistics computed on real, unlabeled inputs; ours does so per evaluation batch rather than through a one-time recalibration pass. Under this protocol, measured accuracy depends on the composition of the evaluation minibatch. Baselines are run and evaluated with their authors' recommended settings (Appendix F).

## 6 Experiments

### 6.1 CIFAR-10/CIFAR100

We evaluate both NormShift-KD instantiations on CIFAR-10 (10 classes) and CIFAR-100 (100 classes): NormShift-KD–BN (RejS + CS mode) on BatchNorm teachers, and NormShift-KD–LN (RejS + Batch Alignment) on LayerNorm teachers obtained by replacing all BN layers in the architecture with LN. RejS is applied with oversample factor $k = 4$. Baselines include Gaussian + RS (BN teachers run in evaluation mode), DDG (Frank & Davis, 2025), and Gaussian + CS (Raikwar & Mishra, 2022). Training details are provided in Appendix F; results are summarized in Tables 3 and 5.

**NormShift-KD–BN: rejection-sampled CS-mode supervision.** NormShift-KD–BN outperforms prior noise-driven baselines across all four teacher-student pairs on CIFAR-10 (Table 3). Running statistics (RS mode) yields near-random performance (13.29% for R34→R18), consistent with the analysis in Section 3. Among CS-mode baselines, Frank & Davis (Frank & Davis, 2025) achieve 65.42% using data-dependent Gaussian noise and Raikwar & Mishra (Raikwar & Mishra, 2022) reach 86.24% with current batch statistics; NormShift-KD–BN achieves 92.44%, a gain of +6.20 points over the strongest noise-based baseline. The

Table 5: Noise-driven data-free distillation accuracy (%) on CIFAR-100. DDG: data-dependent Gaussian.

| Teacher
*Teacher Acc.* | BN Teacher
WRN28-10
*81.02* | | LN Teacher
WRN28-10
*75.98* | |
|---|---|---|---|---|
| **Student** | WRN16-8 | RN18 | WRN16-8 | RN18 |
| Gaussian + RS | 1.21 | 1.26 | 1.26 | 2.27 |
| DDG Frank & Davis (2025) | 20.45 | 24.22 | 1.12 | 1.94 |
| Gaussian + CS Raikwar & Mishra (2022) | 65.70 | 55.38 | 1.28 | 2.25 |
| **NormShift-KD** | **72.12** | **71.04** | **52.78** | **51.25** |

improvement holds across architectures: +6.98 points for R34→MobileNetV2 (89.77% vs. 82.79%), +7.76 points for WRN28-10→WRN16-8 (87.58% vs. 79.82%), and +3.86 points for WRN28-10→R18 (89.17% vs. 85.31%). On CIFAR-100 (Table 5), the gap to data-dependent Gaussian widens: 72.12% vs. 20.45% for WRN28-10→WRN16-8, and 71.04% vs. 24.22% for WRN28-10→R18, indicating that rejection-sampled CS-mode supervision remains effective as the label space grows.

**NormShift-KD–LN: noise-driven distillation from LayerNorm teachers.** Section 3 showed that noise-driven distillation fails when teachers use per-sample normalization: a pure LayerNorm ResNet-34 teacher with 91.40% real-data accuracy yields only 18.00% student accuracy, a failure traceable to the absence of batch-level coupling. NormShift-KD–LN recovers student accuracy to 79.56% for R34→R18 on CIFAR-10 (Table 3), with similar recovery across the other pairs: 75.72% for R34→MobileNetV2 (from 12.82%), 74.42% for WRN28-10→WRN16-8 (from 10.32%), and 79.37% for WRN28-10→R18 (from 10.48%). On CIFAR-100 (Table 5), the failure mode is more pronounced: a WRN28-10-LN teacher with 75.98% accuracy produces 1.26% student accuracy without batch alignment—near-chance on 100 classes despite a well-performing teacher. NormShift-KD–LN recovers 52.78% for WRN28-10→WRN16-8 and 51.25% for WRN28-10→R18, gains of +51.5 and +48.98 points respectively. A gap to NormShift-KD–BN remains (e.g., 72.12% vs. 52.78% on WRN28-10→WRN16-8), but NormShift-KD–LN converts a setting where distillation fails outright into one where useful transfer occurs. To our knowledge, this is the first reported noise-driven distillation from LayerNorm teachers.

## 6.2 Ablation on the mixing coefficient $\alpha$

Table 6 reports an ablation over the mixing coefficient $\alpha \in [0, 1]$ defined in Section 5, on a WRN-28-10-LN teacher and ResNet-18 student on CIFAR-10. All other factors are held fixed: batch size 1024, 400 batches per epoch, RejS oversample factor $K = 4$, temperature $T = 1$, and CS mode.

Table 6: Effect of the mixing coefficient $\alpha$ on student accuracy. WRN-28-10-LN teacher and ResNet-18 student on CIFAR-10. The response is non-monotone, with peak accuracy at $\alpha \in [0.5, 0.75]$ and a $\sim 10$pp drop at the pure-batch-alignment endpoint $\alpha = 1.0$.

| $\alpha$ | 0.00
(pure LN) | 0.25 | 0.50 | 0.75 | 1.00
(pure BA) |
|---|---|---|---|---|---|
| Best val. acc. (%) | 10.45 | 33.08 | **79.79** | 79.37 | 69.63 |

The two endpoints behave as expected. At $\alpha = 0$ the wrapper reduces to ordinary LayerNorm and the student collapses to chance accuracy, confirming that batch alignment is necessary for the LN-teacher setting. At $\alpha = 1$ the wrapper fully replaces LN with batch-level statistics, recovering most but not all of the available accuracy. The interior of the range is where the response is most informative: $\alpha \in [0.5, 0.75]$ improves over $\alpha = 1$ by roughly 10pp, indicating that the LN affine parameters $(\gamma, \beta)$ retain useful information that pure batch alignment discards. The two normalizations are complementary rather than substitutable. LN's learned parameters reflect per-sample adaptation acquired during the teacher's real-data training, while batch

alignment supplies the inter-sample coupling that noise-driven distillation requires; mixing them preserves both signals.

We use $\alpha = 0.5$ as the default in subsequent experiments. We treat this recommendation as a preliminary basis for selecting $\alpha$ on similar teacher-student pairs rather than a universal default: the optimum may shift with class count, teacher capacity, or input dimensionality, and a fuller characterization is left for future work. Appendix C reports a teacher-side diagnostic based on the mutual information $I(X; C)$ between Gaussian inputs and teacher pseudo-labels, whose argmax over $\alpha$ matches the student accuracy peak in this ablation and is cheap to compute without training the student.

Rejection sampling ablation is shown in Appendix C.1.

### 6.3  Scaling to ImageNet

We evaluate NormShift-KD on ImageNet-scale data to assess how both instantiations behave as image resolution and number of classes increase. Experiment details are in Appendix F. We employ ResNet-34 as the teacher for all experiments. Models are initialized from ImageNet-1K pretrained weights and fully fine-tuned on each subset using a low learning rate.

Table 7: Knowledge distillation results on ImageNet subsets. Student: ResNet-18 distilled from a ResNet-34 teacher. Noise rows use NormShift-KD's BN or LN instantiation as indicated.

| Dataset | Teacher Norm | Method | Teacher Acc. | Student Acc. |
|---|---|---|---|---|
| ImageNet-10 | BN | Real-Data
NormShift-KD–BN | 86.20 | 65.08
**75.79** |
| | LN | Gaussian (no Batch Align.)
NormShift-KD–LN | 85.04 | 11.82
**69.72** |
| ImageNet-100 | BN | Real-Data
NormShift-KD–BN | 85.60 | **80.94**
68.54 |
| | LN | Gaussian (no Batch Align.)
NormShift-KD–LN | 84.51 | 1.84
**50.64** |

Table 7 summarizes our main results. On ImageNet-10, NormShift-KD–BN outperforms real-data KD by a large margin (75.79% vs. 65.08%). This setting is small (5,000 training samples), which limits the diversity available to real-data KD and makes vanilla KD prone to overfitting when only 500 samples per class are used. In contrast, noise-driven distillation can draw from an effectively unbounded input space while still exploiting the teacher's BatchNorm-induced batch coupling under CS mode. All distillation runs use the same standard augmentation pipeline used for real-data KD. On ImageNet-100, the trend reverses: real-data KD reaches 80.94% whereas NormShift-KD–BN attains 68.54%. ImageNet-100 provides substantially more supervision (130k training samples, i.e., 1,300 per class), which strengthens real-data KD. Moreover, at this scale noise-based KD becomes more sensitive to the stability of batch-statistics-driven supervision; in our current setup the effective batch size is constrained by available GPU resources, which may further limit performance.

The LayerNorm-teacher setting follows the same qualitative pattern as on CIFAR. Without batch alignment, the LN teacher yields near-chance student accuracy at both scales (11.82% on ImageNet-10, 1.84% on ImageNet-100). NormShift-KD–LN recovers these to 69.72% and 50.64% respectively. A gap to NormShift-KD–BN remains at both scales (6.07 points on ImageNet-10, 17.90 on ImageNet-100), consistent with the residual sharpness gap analyzed in Appendix C: batch-aligned LN teachers recover class coverage but not the per-sample confidence that BN's running statistics impart during training. These results extend the CIFAR-scale finding that NormShift-KD–LN converts an outright failure regime into useful transfer, and indicate that the framework continues to deliver non-trivial accuracy as resolution and class count grow.

NormShift-KD is most competitive in data-scarce settings (ImageNet-10), where unbounded synthetic inputs compensate for limited real diversity. With substantially more real supervision (ImageNet-100), real-data KD remains stronger for BN teachers, while NormShift-KD–LN remains the only working option for LayerNorm teachers in the noise-only regime.

The class-imbalance analysis of Section 3.1 (Table 1) explains why rejection sampling becomes computationally prohibitive at ImageNet-1K scale.

# 7 Limitations and Future Work

## 7.1 Attention Collapse in Vision Transformers

NormShift-KD successfully addresses normalization-induced failures in convolutional architectures. Vision Transformers, however, face an orthogonal challenge: the self-attention mechanism itself degrades under structureless noise, independent of normalization. We provide a formal characterization of this failure mode.

Consider a single attention head processing tokens $x_1, \ldots, x_N \overset{\text{i.i.d.}}{\sim} \mathcal{N}(0, \sigma^2 I_D)$, with queries $q_i = W_Q x_i$, keys $k_j = W_K x_j$, and attention scores $s_{ij} = q_i^\top k_j / \sqrt{d_k}$.

**Lemma 7.1** (Score Statistics). *Let $A := W_Q^\top W_K$. For $i \neq j$, $\mathbb{E}[s_{ij}] = 0$ and $\text{Var}(s_{ij}) = \sigma_s^2$, where $\sigma_s^2 := \sigma^4 \|A\|_F^2 / d_k$. Conditional on $x_j$, $s_{ij} \mid x_j \sim \mathcal{N}(0, \sigma^2 \|Ax_j\|^2 / d_k)$.*

The score spread $\Delta_i := \max_t s_{it} - \min_t s_{it}$ determines how sharply attention concentrates:

**Theorem 7.2** (Attention Concentration Under Gaussian Noise). *Attention weights satisfy $e^{-\Delta_i}/N \leq a_{ij} \leq e^{\Delta_i}/N$ for all $j$. Under Assumption in Appendix B, with probability at least $1 - \delta$:*

$$\Delta_i \leq 2\sigma_s \sqrt{2 \log(2N/\delta)}. \tag{6}$$

*When $\Delta_i$ is small, attention weights concentrate near uniform: $a_{ij} \approx 1/N$.*

The implications for distillation are immediate. When attention is near-uniform, the output reduces to $o_i \approx \frac{1}{N} \sum_{j=1}^{N} v_j$—independent of the query index $i$. Propagated through multiple layers, this averaging behavior causes different spatial positions to produce increasingly similar representations. The resulting teacher outputs exhibit low diversity across noise batches, yielding high-entropy, weakly discriminative soft targets that provide minimal supervision signal.

We test the prediction of Theorem 7.2 directly on two pretrained transformer teachers (ViT-S, 92.02% top-1 on CIFAR-100; Swin-T, 89.26%) by recording the pre-softmax score spread $\Delta_i = \max_t s_{it} - \min_t s_{it}$ and the post-softmax attention entropy $H(A_q) = -\sum_t a_{q,t} \log a_{q,t}$ at every block, on 256 real CIFAR-100 images and 256 Gaussian noise samples per condition. The theorem's primary prediction is that $\Delta_i$ should be smaller under noise: on ViT-S, the block-averaged gap is $+2.46$ in the predicted direction, with the same ordering at every individual block. Aggregate attention entropy is correspondingly higher under noise on both architectures, again in the predicted direction. The effect is sharpest in early layers: at block 0 of ViT-S, attention entropy under noise drops to 2.63 nats against 3.77 on real images, corresponding to roughly 14 versus 43 effective key positions out of 196 patches. A complementary diagnostic on patch-similarity structure shows the same pattern, with per-block gaps positive at every block of both architectures. Full tables and per-block detail are in Appendix B.4.

The failure mode characterized above is the first of three compounding causes of ViT failure under noise-driven distillation; the other two operate after normalization and after attention, and are not addressed by NormShift-KD. The normalization barrier, which batch alignment addresses, means LN-equipped teachers see no batch coupling at the layer level. Attention collapse, documented empirically above, means the self-attention operator itself fails to preserve informative token distinctions on out-of-distribution input, regardless of whether layer outputs were batch-aligned upstream. The third cause is softmax-space class concentration: even when the first two are mitigated, the teacher's classifier head on Gaussian noise places most of its probability mass on a small subset of classes, which sampling-based mitigations cannot correct (Appendix B.4). NormShift-KD's batch-alignment component addresses the normalization barrier cleanly

and is sufficient for CNN-LN teachers; for ViT-family teachers, the remaining causes appear to require either structured input generation or architectural intervention, which we leave to future work. Full proofs and additional analysis, including the role of self-bias and positional embeddings, are provided in Appendix B.

**Task scope**   Our study focuses exclusively on image classification, leaving open the question of whether noise-driven distillation and NormShift-KD extend to other vision tasks such as object detection, semantic segmentation, and instance segmentation, where spatial structure in teacher outputs is more complex. Additionally, while our method substantially improves distillation from LayerNorm teachers, a gap remains compared to BatchNorm teachers, and Vision Transformers present fundamental challenges—attention collapse under structureless noise—that batch alignment alone cannot address. The inherent class imbalance in teacher predictions over Gaussian inputs (Table 1) limits scalability to ImageNet-1K without prohibitive rejection sampling.

**Future work**   Three directions follow from the limitations above: (i) structured noise generation tailored to attention mechanisms, to address the ViT failure mode that batch alignment cannot reach; (ii) extension to dense prediction tasks and to incremental/continual learning settings where data-free constraints are particularly relevant; (iii) investigate whether similar normalization-aware techniques can enable distillation across detection and segmentation networks.

## 8   Conclusion

We presented a systematic study of noise-driven data-free knowledge distillation, identifying normalization as the factor governing its success. BatchNorm teachers in CS mode yield informative but class-imbalanced supervision on Gaussian inputs; LayerNorm and GroupNorm teachers fail more severely, since per-sample normalization eliminates batch coupling. We introduce NormShift-KD, a normalization-aware framework with two instantiations sharing a rejection-sampling component: NormShift-KD–BN combines RejS with CS-mode inference, and NormShift-KD–LN combines RejS with a batch-alignment wrapper that restores inter-sample coupling without retraining or architectural changes. On CIFAR-10/100, NormShift-KD–BN improves over the strongest noise-driven baselines, and NormShift-KD–LN yields the first reported noise-driven distillation from LayerNorm teachers without any data synthesis. We further analyze why batch-invariant students face irreducible approximation error from batch-conditional teachers and why Vision Transformers suffer attention collapse under structureless noise.

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

## A  Sensitivity to Design Choices

Section 3 established that batch statistics are central to noise-driven distillation. We now examine sensitivity to other design choices: batch size, temperature, and noise type. All experiments use a ResNet-34 teacher and ResNet-18 student on CIFAR-10; results are shown in Figure 2.

**Batch size.**  Larger batches yield consistent gains, improving accuracy from 88.47% at batch size 128 to 91.53% at 4096. This strong dependence reinforces the importance of batch statistics: larger batches yield more stable per-channel estimates, producing more consistent teacher supervision.

**Temperature.**  Moderate temperatures ($\tau$=4) slightly outperform hard targets ($\tau$=1), while excessive softening ($\tau$=8) degrades performance as predictions become nearly uniform. The effect is modest compared to batch size and noise type.

**Noise type.**  Structured noise substantially outperforms unstructured alternatives. Dead Leaves achieves 90.41%, nearly 9% above Perlin noise (81.43%), confirming that edge-like spatial structure activates discriminative teacher features more effectively than smooth, low-frequency patterns.

The strong batch-size dependence we observe reinforces the findings of Raikwar & Mishra (2022), who identified BatchNorm as critical for noise-driven distillation. We extend this analysis in the next section by systematically ablating BatchNorm to quantify how batch coupling affects transfer quality.

## B  Vision Transformer Analysis: Supporting Results and Proofs

This appendix provides complete proofs and supporting results for the attention collapse analysis in Section 7.1.

### B.1  Supporting Lemma

**Lemma B.1** (Exchangeability Implies Uniform Expected Attention). *Fix a query index $i$. If the score vector $(s_{i1}, \ldots, s_{iN})$ is exchangeable across $j$, then $\mathbb{E}[a_{ij}] = 1/N$ for all $j$.*

*Proof.* Exchangeability implies $\mathbb{E}[a_{i1}] = \cdots = \mathbb{E}[a_{iN}]$. Since $\sum_{j=1}^{N} a_{ij} = 1$, we have $\sum_{j=1}^{N} \mathbb{E}[a_{ij}] = 1$, so each expectation equals $1/N$. □

### B.2  Proofs of Main Results

The proof of Theorem 7.2 uses a conditional Gaussianity property of the off-diagonal attention scores. This property is slightly weaker than marginal sub-Gaussianity, but it is sufficient once the conditional variances are controlled. We first restate Lemma 7.1 in the form used by the proof.

**Lemma 7.1 (restated, full form).**  *Let $A := W_Q^\top W_K$, and define*

$$s_{ij} := \frac{x_i^\top A x_j}{\sqrt{d_k}}, \qquad x_1, \ldots, x_N \overset{\text{i.i.d.}}{\sim} \mathcal{N}(0, \sigma^2 I).$$

*For $i \neq j$,*

$$\mathbb{E}[s_{ij}] = 0, \qquad \mathrm{Var}(s_{ij}) = \frac{\sigma^4 \|A\|_F^2}{d_k}.$$

*Moreover, conditional on $x_j$,*

$$s_{ij} \mid x_j \sim \mathcal{N}\left(0, \frac{\sigma^2 \|A x_j\|^2}{d_k}\right).$$

*For the diagonal score,*

$$\mathbb{E}[s_{ii}] = \mu_{\text{self}} := \frac{\sigma^2}{\sqrt{d_k}} \operatorname{tr}(A).$$

*The unconditional off-diagonal variance is denoted $\sigma_s^2 := \sigma^4 \|A\|_F^2 / d_k$.*

*Proof of Lemma 7.1.* For $i \neq j$, independence and $\mathbb{E}[x_i] = \mathbb{E}[x_j] = 0$ give

$$\mathbb{E}[s_{ij}] = \frac{1}{\sqrt{d_k}} \mathbb{E}[x_i]^\top A \, \mathbb{E}[x_j] = 0.$$

For the variance,

$$\mathbb{E}\big[(x_i^\top A x_j)^2\big] = \mathbb{E}\big[x_i^\top A x_j x_j^\top A^\top x_i\big] \tag{7}$$
$$= \mathbb{E}\big[x_i^\top A(\sigma^2 I) A^\top x_i\big] \tag{8}$$
$$= \sigma^2 \operatorname{tr}\big(A A^\top \mathbb{E}[x_i x_i^\top]\big) \tag{9}$$
$$= \sigma^4 \operatorname{tr}(A A^\top) = \sigma^4 \|A\|_F^2. \tag{10}$$

Dividing by $d_k$ yields $\operatorname{Var}(s_{ij}) = \sigma^4 \|A\|_F^2 / d_k$.

For the conditional law, write $s_{ij} = (A x_j)^\top x_i / \sqrt{d_k}$. Conditional on $x_j$, the vector $A x_j$ is fixed, while $x_i \sim \mathcal{N}(0, \sigma^2 I)$. Therefore $s_{ij} \mid x_j$ is a one-dimensional Gaussian random variable with mean zero and variance

$$\frac{1}{d_k}(A x_j)^\top \mathbb{E}[x_i x_i^\top](A x_j) = \frac{\sigma^2 \|A x_j\|^2}{d_k}.$$

For the diagonal score,

$$\mathbb{E}[s_{ii}] = \frac{1}{\sqrt{d_k}} \mathbb{E}[x_i^\top A x_i] = \frac{1}{\sqrt{d_k}} \operatorname{tr}\big(A \mathbb{E}[x_i x_i^\top]\big) = \frac{\sigma^2}{\sqrt{d_k}} \operatorname{tr}(A).$$

$\square$

**Score-regularity condition.** The proof of Theorem 7.2 below uses two regularity conditions on the fixed query index $i$. These conditions are mild when the singular spectrum of $A$ is not concentrated in a few directions and when the diagonal self-score does not dominate the off-diagonal score scale.

The first condition controls the conditional variances of the off-diagonal scores at the unconditional variance scale:

$$\max_{t \neq i} \frac{\sigma^2 \|A x_t\|^2}{d_k} \leq \sigma_s^2 = \frac{\sigma^4 \|A\|_F^2}{d_k}, \quad \text{equivalently,} \quad \max_{t \neq i} \|A x_t\|^2 \leq \sigma^2 \|A\|_F^2. \tag{A1}$$

The second condition bounds the diagonal self-score relative to the off-diagonal scale. Writing $A_{\text{sym}} := (A + A^\top)/2$,

$$|\mu_{\text{self}}| + C\left( \frac{\sigma^2 \|A_{\text{sym}}\|_F}{\sqrt{d_k}} \sqrt{\log(2/\delta)} + \frac{\sigma^2 \|A_{\text{sym}}\|_{\text{op}}}{\sqrt{d_k}} \log(2/\delta) \right) \leq \sigma_s \sqrt{2 \log(2N/\delta)}, \tag{A2}$$

where $C > 0$ is the universal constant from the Hanson–Wright inequality. Condition (A2) is automatic if self-attention is masked in the analysis; otherwise it requires the diagonal quadratic-form mean and fluctuations to be no larger than the off-diagonal score scale.

*Proof of Theorem 7.2.* Fix a query index $i$, and define

$$M := \max_{1 \leq t \leq N} s_{it}, \qquad m := \min_{1 \leq t \leq N} s_{it}, \qquad \Delta_i := M - m.$$

For any $j$, the attention weight is

$$a_{ij} = \frac{e^{s_{ij}}}{\sum_{t=1}^N e^{s_{it}}}.$$

Since $s_{ij} \leq M$ and $s_{it} \geq m$ for all $t$,

$$a_{ij} \leq \frac{e^M}{N e^m} = \frac{e^{\Delta_i}}{N},$$

and analogously $a_{ij} \geq e^{-\Delta_i}/N$. Therefore $e^{-\Delta_i}/N \leq a_{ij} \leq e^{\Delta_i}/N$.

It remains to bound $\Delta_i$. Since $\Delta_i \leq 2\max_t |s_{it}|$, it suffices to control $\max_t |s_{it}|$.

We first control the off-diagonal scores. Condition on the keys $\{x_t : t \neq i\}$. By the conditional Gaussianity established above,

$$s_{it} \mid x_t \sim \mathcal{N}\left(0, \frac{\sigma^2\|Ax_t\|^2}{d_k}\right), \qquad t \neq i.$$

Under condition (A1), each conditional variance is at most $\sigma_s^2$, so the standard Gaussian tail bound gives

$$\Pr\left(|s_{it}| > u \mid x_t\right) \leq 2\exp\left(-\frac{u^2}{2\sigma_s^2}\right).$$

A union bound over the $N-1$ off-diagonal scores and the choice $u = \sigma_s\sqrt{2\log(2N/\delta)}$ yield

$$\max_{t\neq i} |s_{it}| \ \leq \ \sigma_s\sqrt{2\log(2N/\delta)} \tag{11}$$

with probability at least $1-\delta$.

We now control the diagonal score. Since $x_i^\top A x_i = x_i^\top A_{\mathrm{sym}} x_i$, the diagonal score is a Gaussian quadratic form. By the Hanson–Wright inequality, with probability at least $1-\delta$,

$$|s_{ii} - \mu_{\mathrm{self}}| \leq C\left(\frac{\sigma^2\|A_{\mathrm{sym}}\|_F}{\sqrt{d_k}}\sqrt{\log(2/\delta)} + \frac{\sigma^2\|A_{\mathrm{sym}}\|_{\mathrm{op}}}{\sqrt{d_k}}\log(2/\delta)\right).$$

Condition (A2) therefore gives

$$|s_{ii}| \ \leq \ \sigma_s\sqrt{2\log(2N/\delta)} \tag{12}$$

with probability at least $1-\delta$.

Combining (11) and (12) (with the failure probabilities rescaled by $\delta/2$ if needed), with probability at least $1-\delta$,

$$\max_{1\leq t\leq N} |s_{it}| \leq \sigma_s\sqrt{2\log(2N/\delta)},$$

and consequently

$$\Delta_i \leq 2\max_t |s_{it}| \leq 2\sigma_s\sqrt{2\log(2N/\delta)}.$$

$\square$

**Why the regularity conditions are needed.** The off-diagonal score $s_{ij} = x_i^\top A x_j/\sqrt{d_k}$ is a bilinear form in two independent Gaussian vectors, which is not sub-Gaussian in general. Conditional on $x_j$, it is Gaussian with variance $\sigma^2\|Ax_j\|^2/d_k$, and condition (A1) ensures these conditional variances are controlled at the scale $\sigma_s^2$. The diagonal score $s_{ii}$ is different: it is a Gaussian quadratic form with nonzero mean $\mu_{\mathrm{self}}$, and condition (A2) rules out the case where the self-score dominates the row. Without these conditions, a large diagonal score or an unusually large key activation would yield self-attention concentration rather than near-uniform attention.

**Unconditional variant.** Without condition (A1), the same conditional Gaussian argument yields a slightly weaker bound involving $\kappa_i := \max_{t\neq i}\|Ax_t\|^2/(\sigma^2\|A\|_F^2)$. Conditioned on the keys, with probability at least $1-\delta$,

$$\max_{t\neq i} |s_{it}| \leq \sigma_s\sqrt{2\kappa_i\log(2N/\delta)}.$$

The theorem's stated bound corresponds to the regular regime $\kappa_i \leq 1$ together with negligible diagonal self-bias. A fully unconditional statement can instead bound $\kappa_i$ using Hanson–Wright, introducing additional terms in $\|A\|_{\mathrm{op}}$, $\|A^\top A\|_F$, and $\log(N/\delta)$.

### B.3 Additional Remarks

*Remark* B.2 (When Self-Bias is Negligible). Lemma 7.1 shows that diagonal scores have mean $\mathbb{E}[s_{ii}] = (\sigma^2/\sqrt{d_k})\,\mathrm{tr}(W_Q^\top W_K)$. Self-bias is negligible when

$$\frac{|\mathrm{tr}(W_Q^\top W_K)|}{\|W_Q^\top W_K\|_F} \ll \sqrt{d_k}. \tag{13}$$

This condition holds when $W_Q$ and $W_K$ are approximately orthogonal or when $d_k$ is large. When self-bias is significant ($\mathbb{E}[s_{ii}] \gg \sigma_s$), each token attends primarily to itself, producing nearly diagonal attention—a different failure mode that partially preserves token identity but severely limits cross-token interaction.

*Remark* B.3 (Positional Embeddings). Positional embeddings break the exact exchangeability assumed in Lemma B.1, since $x_i \mapsto x_i + p_i$ makes tokens non-identically distributed. However, positional embeddings do not introduce semantic correlations between token content and position under i.i.d. noise inputs. The score spread $\Delta_i$ may increase due to position-dependent biases, but attention typically remains high-entropy when token content is structureless. Empirically, we observe that positional embeddings provide insufficient structure to overcome the fundamental uniformity induced by random token content.

*Remark* B.4 (Connection to Representation Collapse). The uniform attention phenomenon connects to broader observations about representation collapse in Transformers. When $a_{ij} \approx 1/N$, each attention layer acts as a spatial averaging operator: $o_i \approx \frac{1}{N}\sum_{j=1}^{N} v_j$ for all query positions $i$. Composed across $L$ layers, this averaging compounds, progressively homogenizing token representations. The [CLS] token, which aggregates information for classification, receives an averaged representation that lacks the discriminative spatial structure present in representations computed on natural images. This explains why ViT teachers produce low-diversity predictions on noise batches despite achieving high accuracy on real data.

### B.4 Empirical evidence for attention collapse on ViT-S and Swin-T

Theorem 7.2 predicts that under i.i.d. Gaussian inputs, the score spread $\Delta_i = \max_t s_{it} - \min_t s_{it}$ is bounded with high probability, and that small $\Delta_i$ forces the attention weights $a_{ij}$ toward the uniform distribution $1/N$. We test this prediction directly on two pretrained transformer teachers, by recording pre-softmax scores and post-softmax attention weights at every block under both real CIFAR-100 images and Gaussian noise, and we report a downstream consequence on patch-similarity entropy as supporting evidence.

**Setup.** We extend the analysis from the ViT-Tiny architecture used in Table 4 to two larger pretrained transformer teachers, ViT-S (92.02% top-1 on CIFAR-100) and Swin-T (89.26% top-1 on CIFAR-100), to characterize the failure mechanism across a range of ViT-family architectures. For each architecture we forward 256 real CIFAR-100 test images and 256 Gaussian noise samples at $224 \times 224$, matched to the per-channel statistics of the validation transform. We patch the attention forward pass to materialize the full pre-softmax score matrix $s = QK^\top/\sqrt{d_k}$ and the post-softmax attention matrix $A$ at every block and head, rather than relying on the fused-attention call that does not expose them. From these we record per-(block, head, query) values of $\Delta_i$ and $H(A_q) = -\sum_t a_{q,t} \log a_{q,t}$. For ViT-S, attention metrics are computed over patch-to-patch attention only, excluding the [CLS] token from both query and key positions, since [CLS]-token attention reflects a specialized aggregation function rather than the patch-level dynamics that Theorem 7.2 addresses.

**Score spread $\Delta_i$ on ViT-S.** Table 8 reports the block-averaged score spread on ViT-S. Noise produces smaller score spread than real images at the aggregate level, with a gap of $+2.46$ in the direction predicted by Theorem 7.2, and the same ordering holds at every individual block of the network. This is the cleanest direct test of the theorem in the appendix, since $\Delta_i$ is the exact quantity the theorem bounds.

Both architectures show higher mean attention entropy under Gaussian noise than under real images, in the direction predicted by Theorem 7.2. On ViT-S the patch-only aggregate gap is $-0.050$ nats (real $-$ noise), and on Swin-T the corresponding gap is $-0.107$ nats. The magnitudes are modest at the aggregate level, but the direction is consistent across both architectures, and the per-block detail in the next paragraph shows where the effect is concentrated.

Table 8: Block-averaged pre-softmax score spread $\Delta_i = \max_t s_{it} - \min_t s_{it}$ on ViT-S, computed over 256 inputs per condition with the [CLS] token excluded. Noise produces smaller score spread than real images, in the direction predicted by Theorem 7.2.

| Condition | Mean $\Delta_i$ | Std $\Delta_i$ | Gap (real $-$ noise) |
|---|---|---|---|
| Gaussian noise | $\approx 8.0$ | $\approx 1.5$ | |
| Real CIFAR-100 | $\approx 10.5$ | $\approx 2.8$ | $+2.46$ |

At the first transformer block of ViT-S, attention entropy under Gaussian noise drops to $H(A_q) = 2.63$ nats against $H(A_q) = 3.77$ nats on real images, a gap of $+1.15$ nats with the maximum value $\log N$ corresponding to the patch token count of 196. The exponentiated entropy reads as the effective number of key positions receiving non-trivial attention mass: under real images this is roughly $e^{3.77} \approx 43$ keys, while under Gaussian noise it is roughly $e^{2.63} \approx 14$. This is a clear instance of attention collapse in the direction the theorem predicts, with the noise distribution concentrating on roughly a third as many keys as the real-image distribution.

The strength of the attention-collapse signature is depth-dependent. Early blocks show the theorem-predicted compression most cleanly, with the largest entropy gap and the sharpest concentration on noise. At later blocks (8–11 on ViT-S) the relationship inverts in entropy: real-image attention becomes more concentrated than noise attention, with gaps of $+0.38$ to $+0.60$ nats. This inversion is not an artifact of the [CLS] token (it persists with [CLS] excluded) and reflects a property of the trained network. Real images drive late-layer attention to develop the concentrated, structured patterns the network was trained to converge on, while noise inputs leave attention diffuse at those layers because the network has no learned response to them. Both patterns are departures from the trained regime; the early blocks show the regime predicted by Theorem 7.2 most directly, while the late blocks show a complementary signature in which real images recruit learned attention structure that noise does not.

We complement the direct attention measurements with a downstream diagnostic that captures how heterogeneous patch representations remain through the network. At the output of each block $l$, we form the cosine patch-similarity matrix $\Gamma_l[i,j] = \cos(z_l[i], z_l[j])$ over patch tokens, take its off-diagonal entries as a one-dimensional distribution on $[-1, 1]$, and bin them into a histogram with $n_b$ uniform bins. We report the discrete histogram entropy $H_l = -\sum_i p_i \log p_i$, summed across blocks: $L_{\text{PSE}} := \sum_l H_l$. Higher $L_{\text{PSE}}$ corresponds to a richer mix of patch-pair similarities through the network; complete homogenization of token representations concentrates similarities near 1 and lowers $H_l$. Histogram-based estimators of similarity-distribution entropy of this form have been used in prior work on transformer analysis Li et al. (2023).

The metric is bin-count-dependent in absolute terms but the gap between conditions is bin-invariant; we verified this by sweeping bin counts across $\{32, 64, 128\}$ and observing that the real-versus-noise gap changes by less than 1.5% (Table 9).

Table 9: Discrete patch-similarity entropy $L_{\text{PSE}}$ summed over blocks, with bin-count sensitivity. Real-image $L_{\text{PSE}}$ exceeds noise $L_{\text{PSE}}$ on both architectures across all bin counts; the gap is bin-invariant within 1.5%.

| Architecture | Condition | bins$= 32$ | bins$= 64$ | bins$= 128$ |
|---|---|---|---|---|
| Swin-T | Gaussian noise | $17.10 \pm 1.89$ | $24.44 \pm 2.12$ | $32.42 \pm 2.24$ |
| Swin-T | Real CIFAR-100 | $25.59 \pm 1.41$ | $33.64 \pm 1.42$ | $41.83 \pm 1.43$ |
| | *Gap* | *+8.49* | *+9.20* | *+9.42* |
| ViT-S | Gaussian noise | $24.07 \pm 0.37$ | $32.17 \pm 0.37$ | $40.42 \pm 0.37$ |
| ViT-S | Real CIFAR-100 | $28.42 \pm 1.01$ | $36.62 \pm 1.02$ | $44.90 \pm 1.02$ |
| | *Gap* | *+4.35* | *+4.45* | *+4.48* |

The per-block $L_{\text{PSE}}$ gap is positive at every block of both architectures (12 of 12 on Swin-T, 12 of 12 on ViT-S). On Swin-T the largest per-block gap is $+1.76$ nats at block 4, the start of stage 2; on ViT-S the largest gap is $+0.91$ nats at block 0, with the gap narrowing approximately monotonically through depth. This pattern matches the score-spread and attention-entropy measurements above: the strongest signature of

attention collapse appears in early-to-middle blocks, and the magnitude of the effect is larger on Swin-T than on ViT-S.

**Implications for ViT-Tiny in Table 4.** Batch alignment intervenes at LayerNorm outputs and restores the batch coupling that LN otherwise discards. The wrapper sits before the attention operator, while the failure documented here occurs inside attention: score spreads compress, early-block attention collapses to a small effective set of keys, and patch representations through the network differ markedly from the real-image regime in ways no upstream normalization fix can repair. The student receives soft targets shaped by this collapsed attention regime, and the resulting supervision is too weak to learn from. This is consistent with the 14% accuracy reported for ViT-Tiny in Table 4, and with the scope claim that NormShift-KD's batch-alignment component addresses the normalization barrier but not the attention failure.

## C   Teacher-side diagnostic for selecting $\alpha$

This appendix reports a teacher-side diagnostic that tracks the alpha response observed in Section 6.2 and motivates the default $\alpha = 0.5$ recommended there. The diagnostic is preliminary: it was developed on a single teacher-student configuration on CIFAR-10, and its generalization to other datasets, architectures, and class counts is not verified here.

**Setup.** For a fixed teacher $T$ operating in CS mode on Gaussian noise inputs $x \sim \mathcal{N}(0, I)$, let $p_T(\cdot \mid x)$ denote the teacher's softmax output and $\bar{P}(\cdot) = \mathbb{E}_x[p_T(\cdot \mid x)]$ the marginal class distribution over a batch. The mutual information between the input and the teacher's pseudo-label is

$$I(X; C) = H(\bar{P}) - \mathbb{E}_x\big[H(p_T(\cdot \mid x))\big], \tag{14}$$

where $H(\cdot)$ denotes Shannon entropy Shannon (1948). The first term measures how broadly the teacher's predictions cover the class space across the batch; the second measures how sharp each individual prediction is. $I(X; C)$ is large only when both conditions hold: the batch is class-diverse *and* each input receives a confident prediction. Both conditions are required for the teacher to provide a useful supervisory signal in noise-driven distillation.

We compute $I(X; C)$ for the same WRN-28-10-LN teacher and the five values of $\alpha$ in the body ablation, with a BN-CS teacher included as a reference. Each measurement uses batch size 1024 and averages over 4 independent Gaussian noise batches. The estimator for $H(p_T(\cdot \mid x))$ uses the closed-form softmax entropy; $H(\bar{P})$ is computed from the empirical marginal over the batch.

Table 10: Mutual information $I(X; C)$ between Gaussian inputs and teacher pseudo-labels, decomposed into the batch marginal entropy $H(\bar{P})$ and the mean per-sample entropy $\mathbb{E}[H(p)]$. WRN-28-10 teachers on CIFAR-10, batch size 1024, CS mode. The student accuracy from the body ablation is repeated for reference.

| Teacher / $\alpha$ | $H(\bar{P})$ | $\mathbb{E}[H(p)]$ | $I(X; C)$ | Student acc. (%) |
|---|---|---|---|---|
| LN+BA, $\alpha = 0.00$ | 1.022 | 0.990 | 0.033 | 10.45 |
| LN+BA, $\alpha = 0.25$ | 1.564 | 1.446 | 0.118 | 33.08 |
| LN+BA, $\alpha = 0.50$ | 2.262 | 2.032 | **0.229** | **79.79** |
| LN+BA, $\alpha = 0.75$ | 2.267 | 2.134 | 0.133 | 79.37 |
| LN+BA, $\alpha = 1.00$ | 2.294 | 2.238 | 0.056 | 69.63 |
| BN-CS (reference) | 2.292 | 0.954 | 1.338 | — |

The mutual information $I(X; C)$ peaks at $\alpha = 0.5$ with the same ordering as the student accuracy across the sweep ($0.5 > 0.75 > 1.0 > 0.25 > 0$). The peak corresponds to a teacher that simultaneously achieves broad class coverage on the batch ($H(\bar{P}) = 2.262$, close to the BN-CS reference value of 2.292) and the sharpest per-sample distributions of any LN+BA configuration ($\mathbb{E}[H(p)] = 2.032$, lower than every higher $\alpha$). The two endpoints fail for distinct reasons that the decomposition makes visible. At $\alpha = 0$ the teacher collapses to a small subset of classes ($H(\bar{P}) = 1.022$), so the supervision contains little class information regardless of how sharp individual predictions are. At $\alpha = 1$ the batch marginal matches BN-CS, but the per-sample entropy is

at its highest in the sweep ($\mathbb{E}[H(p)] = 2.238$), so individual inputs do not carry strong class commitments. The interior point $\alpha = 0.5$ balances both terms and yields the highest $I(X; C)$.

At $\alpha = 0.5$, the LN+BA teacher's $I(X; C)$ recovers approximately 17% of the BN-CS reference (0.229 versus 1.338). The gap is driven entirely by per-sample sharpness: the two teachers match on $H(\bar{P})$, but the BN-CS teacher achieves $\mathbb{E}[H(p)] = 0.954$, more than $2\times$ lower than the best LN+BA configuration. Batch alignment can recover the batch-level class distribution that LN otherwise loses, but it does not recover the per-sample confidence that BN's running statistics impart during normal training. This is a structural ceiling on what LN+BA can achieve in the noise-only regime regardless of $\alpha$, and is consistent with the residual gap between LN-teacher and BN-teacher results in Section 6.

Equation 14 requires only forward passes through the frozen teacher on Gaussian noise; no student training is required. Sweeping $\alpha$ on a candidate teacher and selecting the value that maximizes $I(X; C)$ takes minutes per teacher and matched the student-accuracy argmax in the configuration tested here. We use this procedure as a preliminary basis for selecting $\alpha$ on similar teacher-student pairs. Whether the alignment between $I(X; C)$ and student accuracy holds at larger class counts, different architectures, or different input distributions is an open question that we have left for future work.

### C.1 Ablation on rejection sampling

Section 6 introduced rejection sampling (RejS) for class-balancing noise batches before distillation. Table 11 reports a matched-protocol ablation that isolates its contribution. Two runs on a WRN-28-10-BN teacher and ResNet-18 student on CIFAR-100 are identical except for the RejS flag: same batch size, optimizer, schedule, seed, and noise distribution in either run.

Table 11: Effect of rejection sampling on student accuracy. WRN-28-10-BN teacher and ResNet-18 student on CIFAR-100, 200 epochs, batch size 256, identical protocol except for RejS. RejS contributes +5.03pp at convergence and widens to +8.24pp at the mid-training peak.

| Configuration | Acc. @ ep. 50 (%) | Acc. @ ep. 200 (%) |
|---|---|---|
| Gaussian + CS (no RejS) | 33.19 | 55.38 |
| Gaussian + CS + RejS | **41.43** | **60.41** |
| $\Delta$ | +8.24 | +5.03 |

RejS contributes a substantial part of the gain over the CS-mode baseline of Raikwar & Mishra Raikwar & Mishra (2022), while the remaining improvement comes from the larger batch size and other protocol differences.

## D Student Architecture Analysis: Supporting Material

This appendix provides complete experimental details and supporting analysis for the student architecture requirements discussed in Section 5.2.

### D.1 Batch-Conditional vs. Batch-Invariant Representations

We formalize the distinction between batch-conditional and batch-invariant architectures.

When a BatchNorm teacher operates in CS mode, activations are normalized using statistics computed from the current minibatch $\mathcal{B} = \{x_1, \ldots, x_m\}$:

$$\text{BN}(h) = \gamma \cdot \frac{h - \mu_{\mathcal{B}}}{\sigma_{\mathcal{B}}} + \beta,$$
$$\mu_{\mathcal{B}} = \frac{1}{m} \sum_{i=1}^{m} h_i, \quad \sigma_{\mathcal{B}}^2 = \frac{1}{m} \sum_{i=1}^{m} (h_i - \mu_{\mathcal{B}})^2, \tag{15}$$

where $h_i$ denotes the hidden activation for sample $x_i$, and $(\gamma, \beta)$ are learned parameters. This couples the representation of each sample to all others in the batch, making the teacher's output explicitly batch-dependent: $p_T(y \mid x, \mathcal{B}) = \sigma(z_T(x; \mathcal{B})/\tau)$.

In contrast, LayerNorm normalizes each sample independently:

$$\mathrm{LN}(h) = \gamma \cdot \frac{h - \mu_h}{\sigma_h} + \beta,$$

$$\mu_h = \frac{1}{d} \sum_{j=1}^{d} h_j, \quad \sigma_h^2 = \frac{1}{d} \sum_{j=1}^{d} (h_j - \mu_h)^2, \tag{16}$$

where $d$ is the feature dimension. The student output $p_S(y \mid x) = \sigma(z_S(x)/\tau)$ depends only on the individual input—we call this *batch-invariance*.

## D.2 Proof of Proposition 5.1

*Proof.* For a fixed input $x$, the optimization over $p_S$ decomposes as:

$$\mathbb{E}_{\mathcal{B}}[\mathrm{KL}(p_T \| p_S)] = \mathbb{E}_{\mathcal{B}}[H(p_T, p_S)] - \mathbb{E}_{\mathcal{B}}[H(p_T)]. \tag{17}$$

Since the second term is independent of $p_S$, minimizing KL divergence is equivalent to minimizing the cross-entropy $\mathbb{E}_{\mathcal{B}}[H(p_T, p_S)]$. By linearity:

$$\mathbb{E}_{\mathcal{B}}[H(p_T, p_S)] = -\sum_y \mathbb{E}_{\mathcal{B}}[p_T(y \mid x, \mathcal{B})] \log p_S(y \mid x) = H(\bar{p}, p_S), \tag{18}$$

which is minimized when $p_S = \bar{p}$, yielding $H(\bar{p})$. Substituting, the minimum KL divergence is $H(\bar{p}) - \mathbb{E}_{\mathcal{B}}[H(p_T)]$. By Jensen's inequality applied to the concave entropy function, this quantity is non-negative, with equality if and only if $p_T(\cdot \mid x, \mathcal{B})$ is constant in $\mathcal{B}$ almost surely.

The information-theoretic interpretation follows directly:

$$H(\bar{p}) - \mathbb{E}_{\mathcal{B}}[H(p_T)] = I(Y; \mathcal{B} \mid X = x). \tag{19}$$

$\square$

## D.3 Why Batch Coupling Emerges Under Noise

With natural images from the training distribution, samples share similar low-level statistics (mean pixel intensity, local contrast), yielding stable batch statistics $\mu_{\mathcal{B}}$ and $\sigma_{\mathcal{B}}^2$ across minibatches. The teacher's predictions consequently exhibit low variance across batch contexts.

Under Gaussian noise inputs, this stability breaks down. The batch mean $\mu_{\mathcal{B}}$ and variance $\sigma_{\mathcal{B}}^2$ become highly variable across different noise realizations. BatchNorm propagates this randomness through the network, causing the logits $z_T(x; \mathcal{B})$ to fluctuate substantially even when the input $x$ is held fixed. The distillation target for any given input thus becomes a random variable over batch contexts, activating the irreducible error identified in Proposition 5.1.

## D.4 Experimental Protocol: Measuring Batch Dependence

To directly measure batch-dependence in teacher and student outputs, we design the following experiment.

**Protocol.** We fix a single noise input $x_0 \sim \mathcal{N}(0, I)$ and construct 30 different batch contexts $\{\mathcal{B}_1, \ldots, \mathcal{B}_{30}\}$, each containing $x_0$ alongside 127 additional i.i.d. noise samples. For each batch context, we record the model's softmax output for $x_0$ and compute:

- **Argmax flip rate**: The fraction of batch pairs where the predicted class differs.

- **Probability variance**: The average variance of each class probability across contexts.

- **Unique predictions**: The set of distinct predicted classes observed.

**Results.** Table 12 reports quantitative metrics, and Figure 3 visualizes the softmax distributions.

Table 12: Batch coupling diagnostics for a fixed noise input across 30 batch contexts. Only models with batch-conditional outputs (via BatchNorm) can match the teacher's varying predictions.

| Model | Norm | Batch Cond. | Argmax Flip (%) | Prob. Var. | Pred. Classes |
|-------|------|-------------|-----------------|------------|---------------|
| Teacher (ResNet-34) | BN | ✓ | 26.7 | 0.0038 | 3, 5, 8 |
| ConvNeXt | LN | ✗ | 0.0 | 0.0000 | 5 |
| ConvNeXt | BN | ✓ | 40.0 | 0.0058 | 1, 3, 6 |

The BatchNorm teacher exhibits 26.7% argmax flip rate: for the same input $x_0$, it predicts class 3, 5, or 8 depending on the batch context. The LayerNorm student produces identical outputs across all contexts—zero flips, zero variance, a single predicted class—confirming batch-invariance. The BatchNorm student shows 40.0% flip rate and non-zero variance, indicating batch-conditional behavior that enables learning.

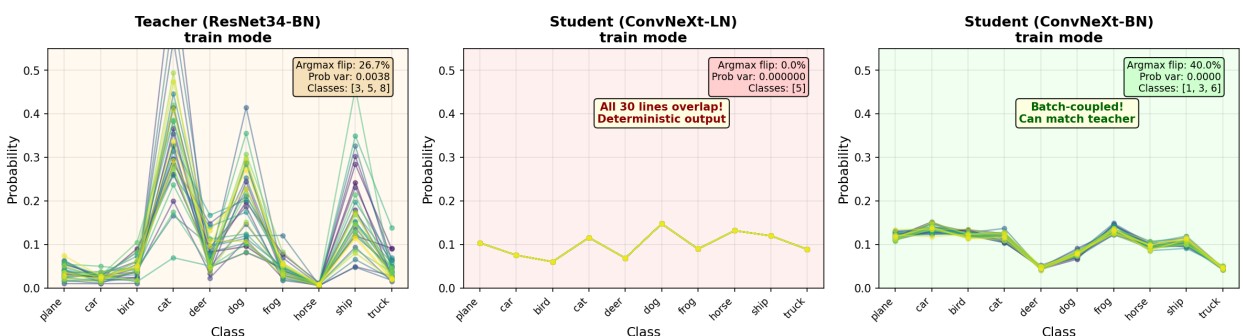

Figure 3: Softmax distributions for a fixed noise input $x_0$ across 30 batch contexts. Each line represents the 10-class probability distribution in one context. **Left**: Teacher (ResNet-34, BN) outputs vary across contexts, with peaks at different classes. **Center**: ConvNeXt-LN produces identical outputs regardless of context. **Right**: ConvNeXt-BN outputs vary, enabling it to match the teacher's batch-conditional behavior.

### D.5 Vision Transformers: Why Dropout Does Not Help

Vision Transformers rely exclusively on LayerNorm, lacking a natural BatchNorm variant. Although ViT architectures include dropout layers that introduce stochasticity, we demonstrate that dropout-induced variation cannot substitute for batch coupling.

**Architecture.** A standard ViT processes images through: (1) patch embedding via linear projection, (2) addition of positional embeddings and a learnable class token, (3) a stack of Transformer blocks containing LayerNorm, multi-head self-attention, and MLP with dropout, and (4) a final LayerNorm followed by a classification head. All normalization uses LayerNorm, making the architecture batch-invariant when dropout is disabled.

We distinguish three types of output variation:

- **Batch coupling** (BatchNorm): Output depends deterministically on batch statistics.

- **Deterministic** (LayerNorm, no dropout): Output is fixed for a given input.

- **Stochastic** (dropout): Output varies randomly, independent of batch context.

For distillation to succeed, the student's variation must be *correlated* with the teacher's batch-dependent targets. Batch coupling satisfies this requirement: teacher and student process the same batch, so both observe identical batch statistics. Dropout-induced stochasticity does not: the random mask $\xi$ is sampled independently of the batch $\mathcal{B}$.

**Experimental validation.**  Table 13 reports results for ViT-Tiny with and without dropout.

Table 13: Batch coupling diagnostics including ViT. Dropout introduces variation (10% flip rate) but this variation is uncorrelated with batch context.

| Model | Norm | Dropout | Variation Type | Argmax Flip (%) |
|---|---|---|---|---|
| Teacher (ResNet-34) | BN | ✗ | Batch-coupled | 26.7 |
| ConvNeXt | LN | ✗ | Deterministic | 0.0 |
| ConvNeXt | BN | ✗ | Batch-coupled | 40.0 |
| ViT-Tiny | LN | ✗ | Deterministic | 0.0 |
| ViT-Tiny | LN | ✓ | Stochastic | 10.0 |

Without dropout, ViT exhibits 0% flip rate—purely deterministic. With dropout enabled, ViT shows 10% flip rate, but this variation is stochastic: across different forward passes with the same batch, ViT produces different outputs due to different dropout masks. This stochasticity is uncorrelated with the teacher's batch-dependent targets.

**Formal analysis.**  For a given batch $\mathcal{B}$, the teacher produces a fixed target $p_T(\cdot \mid x, \mathcal{B})$. The ViT student with dropout produces:

$$p_S(y \mid x, \xi) = \sigma(z_S(x; \xi)/\tau), \tag{20}$$

where $\xi$ denotes the random dropout mask. Since $\xi$ is sampled independently of $\mathcal{B}$, the student has no mechanism to predict which output the teacher will produce. The expected loss:

$$\mathbb{E}_{\mathcal{B}, \xi}\left[\mathrm{KL}(p_T(\cdot \mid x, \mathcal{B})\|p_S(\cdot \mid x, \xi))\right] \tag{21}$$

involves independent random variables $\mathcal{B}$ and $\xi$. The student's variation over $\xi$ cannot reduce the mismatch caused by the teacher's variation over $\mathcal{B}$—if anything, dropout adds noise to the gradient signal.

### D.6   Detailed Results: Restoring Batch Coupling

Table 14 provides complete results for the batch alignment experiments.

Table 14: Test accuracy on CIFAR-10 for LayerNorm students distilled from a ResNet-34 (BatchNorm) teacher using Gaussian noise. CNXt is ConvNext.

| Student | Norm | Batch Align. | Test Acc. (%) |
|---|---|---|---|
| CNXt-Small | LN | ✗ | 10.0 |
| CNXt-Small | LN | ✓ | **82.2** |
| ViT-Tiny | LN | ✗ | 10.0 |
| ViT-Tiny | LN | ✓ | 14.0 |
| ViT-Tiny (cached targets) | LN | ✓ | 17.0 |

**ConvNeXt analysis.**  Batch Alignment transforms failed distillation (10%) into successful transfer (82.2%). Without Batch Alignment, ConvNeXt-LN produces identical probability distributions regardless of batch context; with Batch Alignment, outputs vary across contexts, enabling the student to track the teacher's batch-dependent targets. ConvNeXt with Batch Alignment preserves convolutional spatial hierarchies and couples activations in a geometry consistent with the teacher's BatchNorm pathway.

**ViT analysis.** For ViT, Batch Alignment alone is insufficient, revealing two distinct barriers:

*Barrier 1: Uncorrelated batch sensitivity.* While ViT can be made batch-sensitive through Batch Alignment, the induced logit variations remain uncorrelated with the teacher's batch-conditional variations. Measuring Pearson correlation $\rho$ between teacher and student logit fluctuations across batch contexts, we find $\rho \approx 0$ for ViT versus substantial correlation for ConvNeXt-BA. Without correlation, the student cannot track the teacher's varying targets.

*Barrier 2: Non-transferable representations.* To isolate target inconsistency from representation learning, we train ViT using cached Monte Carlo-averaged targets that remove per-step target switching. Even with stable targets, accuracy remains at 14–17%, indicating that ViT trained on Gaussian noise fails to learn representations transferable to natural images. This suggests that ViT's token-based pipeline ($[B, N, D]$ tensor structure) cannot align with the teacher's spatial feature statistics ($[B, C, H, W]$), whereas ConvNeXt's convolutional inductive bias enables such alignment.

## E   Comparison with generator based baselines

For reference, Table 15 reports the published CIFAR-10 results of representative synthesis-based methods. These methods address the input-synthesis problem and are therefore not baselines for the setting studied here. NormShift-KD trains no auxiliary network and stores no data: the batch-alignment wrapper introduces two additional reductions (mean, variance) per wrapped layer per forward pass and no trainable parameters, and rejection sampling increases teacher-inference cost by a factor of $k = 4$ (forward passes only).

Table 15: CIFAR-10 ResNet-34→ResNet-18 results of representative synthesis-based data-free distillation methods. These methods address the input-synthesis problem and are reported for reference rather than as baselines.

| Method | Auxiliary machinery | Synth. time | Acc. (%) |
|---|---|---|---|
| DAFL (Chen et al., 2019) | trained generator | 2.73 h | 92.22 |
| DeepInversion (Yin et al., 2020) | per-batch inversion with BN-statistic priors | 42.1 h | 93.26 |
| CMI (Fang et al., 2021) | inversion with a contrastive memory of synthesized images | 19.0 h | 94.84 |
| NormShift-KD–BN (ours, noise-only) | none; noise sampled on the fly (not stored) | — | 92.44 |

## F   Training Details

We distill all the models with batch size of 1024 and number of batches 400. For ImageNet-10/100 we use cross entropy loss on pseudo labels. Pseudo labels are the labels given by the teacher model on noise input. Optimizer is Adam and scheduler is ReduceLRonPlateau. The learning rate is 0.001. The gaussian noise is generated on-the-fly and is not stored. For ImageNet10 during evaluation, the val transforms resize to 256 and center crop to 224, so the model is evaluated at 224x224. The training was done on 64x64 images which get upsampled to 224 by the transforms. The baselines are run with their author-recommended hyperparameters and settings.

We construct ImageNet-10 and ImageNet-100 as subsets of ImageNet-1K containing 10 and 100 randomly selected classes, respectively. ImageNet-10 comprises 5,000 training and 500 validation images; ImageNet-100 comprises approximately 130,000 training and 5,000 validation images.

## G   Additional figures

This section includes additional figures.

## H   Algorithm

Algorithm 1 details the forward pass, and Algorithm 2 presents the complete distillation procedure.

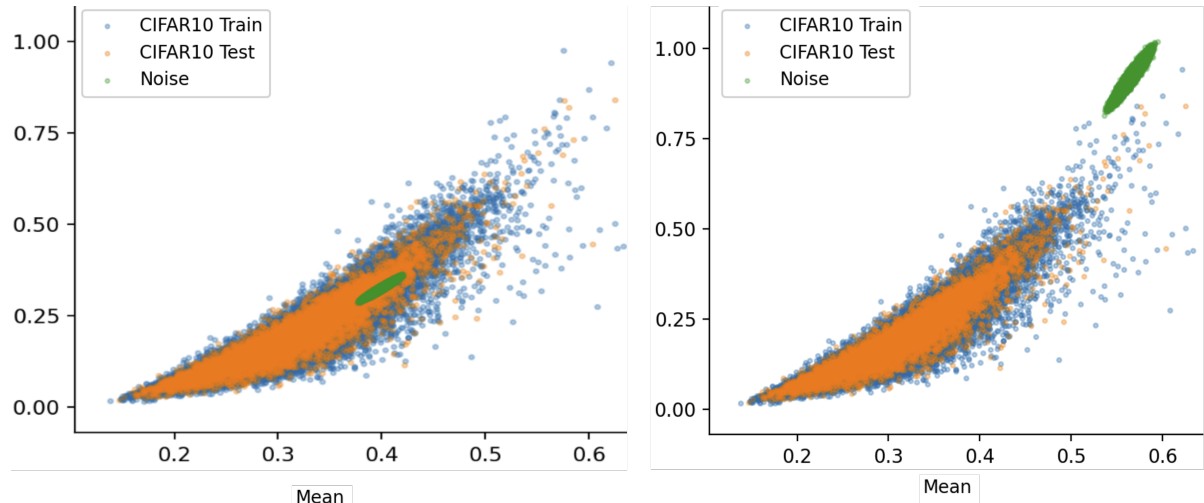

Figure 4: **Activation-statistics mismatch under Gaussian noise depends on BatchNorm mode.** Mean–variance statistics of activations after an early Conv–BN–ReLU block in a ResNet-34 teacher trained on CIFAR-10, computed over CIFAR-10 train (blue), CIFAR-10 test (orange), and Gaussian noise inputs (green). In CS mode (left), BatchNorm uses minibatch statistics computed on the current noise batch, substantially reducing this mismatch and yielding more informative supervision for noise-driven distillation. In RS mode (right), BatchNorm uses running statistics estimated from real images, causing noise-induced activations to occupy a narrow, shifted regime relative to real-data activations.

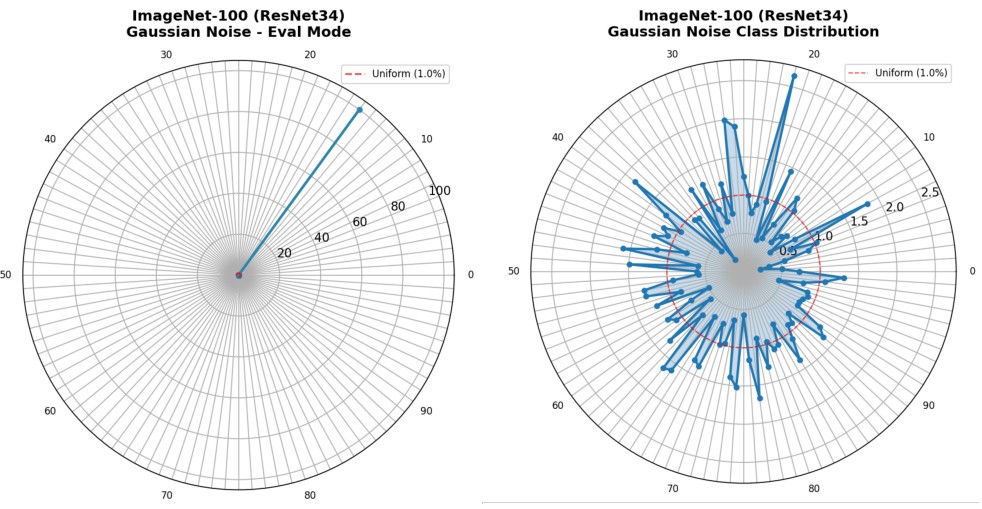

Figure 5: ImageNet-100 class-wise prediction distributions of a ResNet-34 teacher on Gaussian noise, shown for RS mode (a) and CS mode (b).

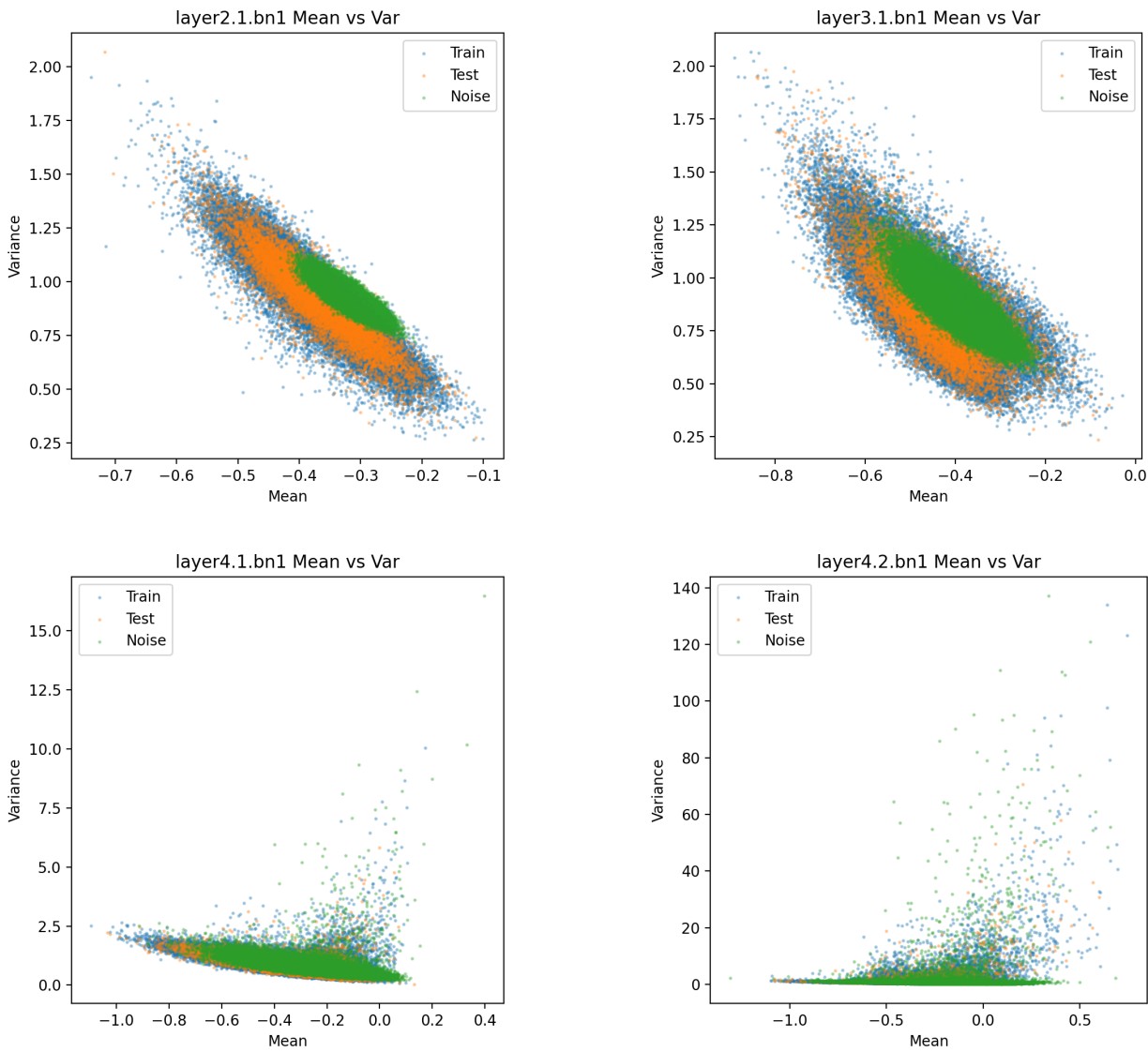

Figure 6: Mean–variance statistics of activations in CS mode after Conv–BN–ReLU block in a ResNet-34 teacher trained on CIFAR-10, computed over CIFAR-10 train (blue), CIFAR-10 test (orange), and Gaussian noise inputs (green).

---

**Algorithm 1** BATCHALIGNEDLN: Forward Pass

---

**Require:** Input $x \in \mathbb{R}^{B \times C \times H \times W}$, learned parameters $\gamma, \beta \in \mathbb{R}^C$, blend factor $\alpha \in [0, 1]$
**Ensure:** Output $y \in \mathbb{R}^{B \times C \times H \times W}$
  1: $\tilde{x} \leftarrow$ LAYERNORM($x$, affine=False) {Per-sample normalization}
  2: **if** $\alpha > 0$ **and** $B > 1$ **then**
  3:     **for** each channel $c \in \{1, \ldots, C\}$ **do**
  4:        $\bar{\mu}_c \leftarrow \frac{1}{BHW} \sum_{b,h,w} \tilde{x}_{b,c,h,w}$
  5:        $\bar{\sigma}_c^2 \leftarrow \frac{1}{BHW} \sum_{b,h,w} (\tilde{x}_{b,c,h,w} - \bar{\mu}_c)^2$
  6:     **end for**
  7:     $\hat{x} \leftarrow (\tilde{x} - \bar{\mu})/\sqrt{\bar{\sigma}^2 + \epsilon}$ {Batch alignment}
  8:     $\tilde{x} \leftarrow (1 - \alpha) \cdot \tilde{x} + \alpha \cdot \hat{x}$ {Soft blend}
  9: **end if**
10: $y \leftarrow \gamma \odot \tilde{x} + \beta$ {Apply learned affine}
11: **return** $y$

---

**Algorithm 2** NormShift-KD–LN: Noise-Driven Distillation from a LayerNorm Teacher

---

**Require:** Pretrained LayerNorm teacher $T$, student $S$ (randomly initialized)
**Require:** Temperature $\tau$, learning rate $\eta$, iterations $N$, batch size $B$, blend factor $\alpha$, RejS oversample factor $k$
**Ensure:** Trained student $S$
  1: Wrap all LayerNorm layers in $T$ with BATCHALIGNEDLN (Alg. 1)
  2: Freeze $T$; set $T$ to CS mode
  3: **for** $i = 1$ to $N$ **do**
  4:     Sample $kB$ inputs from $\mathcal{N}(0, I)^{C_{\text{in}} \times H \times W}$; query $T$ for pseudo-labels; subsample $B$ inputs approximating a uniform pseudo-label distribution (RejS)
  5:     $p_T \leftarrow \text{softmax}(T(x)/\tau)$
  6:     $p_S \leftarrow \text{softmax}(S(x)/\tau)$
  7:     $\mathcal{L} \leftarrow \text{KL}(p_T \,\|\, p_S)$
  8:     $\theta_S \leftarrow \theta_S - \eta \nabla_{\theta_S} \mathcal{L}$
  9: **end for**

---

