# OpenReview forum: "Breaking BatchNorm Barriers for Noise-driven Data Free Knowledge Distillation"
_TMLR — Under review for TMLR_

### Review · Reviewer_hpPg · 2026-06-15

**Summary Of Contributions:**

The paper studies why Gaussian-noise-only data-free knowledge distillation (DFKD) works for BatchNorm models but fails for LayerNorm/GroupNorm models, identifies batch coupling induced by normalization as the key factor, and proposes NormShift-KD, which restores batch-level interactions and substantially improves noise-driven distillation without generators or real data.

**Major Strengths**

Strength 1: This paper asks a clear scientific question: Why does Gaussian-noise KD work at all? This is very novel compared to lots of KD papers where they ask "Here to design trick that improves accuracy."

Strength 2: The BN replacement study is very interesting, which directly tests the hypothesis instead of relying on correlation.

Strength 3: NormShift-KD is remarkably simple: no generator, no diffusion model, no inversion, and no real data. Just: CS mode, rejection sampling, batch alignment. Simpler methods are often more convincing because fewer moving parts exist.

**Major Weaknesses**

Weakness 1: Novelty of the method is somewhat limited. The core insight is strong while the method itself is relatively simple. I suspect this paper is likely to be diagnosis rather than the technique novelty.

Weakness 2: Comparisons are restricted to noise-only methods. The paper compares mainly against: Gaussian+RS, Gaussian+CS, DDG.
But modern DFKD literature contains stronger methods based on: DeepInversion, DAFL, CMI, and so on. The authors intentionally focus on the "noise-only" setting.

Weakness 3: Transformer results remain weak. The paper identifies ViT failure but it does not solve it. But I am okay with it as a limitation.

**Audience:**

Yes

**Audience Explanation:**

This paper addresses a fundamental question in data-free knowledge distillation: why Gaussian-noise-driven distillation succeeds for some architectures but fails for others. The paper provides a systematic analysis showing that normalization choice, particularly the presence of BatchNorm-induced batch coupling, is a key factor that could make the paper published. Beyond the empirical findings, the authors introduce a simple normalization-aware framework (NormShift-KD) that substantially improves noise-driven distillation and, to my knowledge, enables effective distillation from LayerNorm-based teachers for the first time. While the proposed method is relatively simple and its effectiveness remains limited for transformer architectures, the diagnostic insights, controlled ablation studies, and theoretical analysis are likely to be of interest to researchers working on data-free knowledge distillation in the future.

**Broader Impact Concerns:**

No significant broader impact concerns. The work studies data-free knowledge distillation and model compression techniques, with the primary goal of improving understanding and efficiency of knowledge transfer between neural networks. The proposed methods do not introduce new datasets, collect personal information, or create obvious pathways for misuse beyond those already associated with standard model compression and distillation techniques. I do not believe a dedicated broader impact discussion is necessary for my recommendation.

**Claims And Evidence:**

Yes

**Claims Explanation:**

Yes. The submission generally supports its main claims with clear empirical evidence, including comprehensive experiments, ablation studies, and analyses across multiple settings. The presented results are consistent with the stated conclusions, and the methodology is described with sufficient details to allow readers to understand and evaluate the findings. While some limitations and additional analyses could further strengthen the work, the evidence provided is accurate, convincing, and largely sufficient to support the paper's central claims.

**Requested Changes:**

1. Broader comparison with established data-free KD methods. The paper primarily compares against noise-based baselines. While this is justified by the paper's focus, additional discussion or experimental comparison with stronger generator-based data-free KD approaches (DeepInversion, DAFL, CMI, and so on) would help readers better understand the practical competitiveness of NormShift-KD.

2. The paper shows that Vision Transformers remain challenging even after batch alignment. A more detailed discussion of potential future directions for overcoming attention-collapse effects would improve the paper's impact.

Overall, the paper is already technically sound and the above suggestions are intended to strengthen the scope rather than address issues that are necessary for acceptance.

---

### Review · Reviewer_Jt48 · 2026-07-04

**Summary Of Contributions:**

The research studies the data-free KD with gaussian noise without relying on real data or synthesis data.
Some interesting findings (takeaways):
- the normalization(batchnorm/layernorm) plays a critical role
- BN with running stats might collapse but current stats stay smooth
- LN/GN fails due to its batch-irrelevant nature
- NormShift-KD appears reasonable: BN: teacher with current stats and rejection sampling; LN/GN: teacher with additional batch alignment
- ViT fails due to attention collapse -> this might mean the attention mechanism driven LLMs cannot benefit from this work/setting

**Audience:**

Yes

**Audience Explanation:**

Researchers on KD topics/BN/LN architecture studies/transformer mechanisms might be interested.

**Broader Impact Concerns:**

Pros: Data-free KD might work on small models.
Cons:
- The large-scale ImageNet & ViT experiments show that this topic might not work in current "scaling" trend of VLM/LLMs.

**Claims And Evidence:**

Yes

**Claims Explanation:**

The problem itself is interesting. The noise-driven data-free KD is simple but remains unstable before.
Most of the present study lies in the discussion/analysis section. Although the methods like batch alignment and current stats/running stats comparison have been proposed in various KD and DA (domain adaptation) studies, but the present study focuses on the data-free KD scenario and indeed discusses well with analysis. Experiments on small and large imagenet datasets also shed light on the generalization warning, which is that the large-scale KD should still be carried out on real/synthetic ones with semantics.

**Requested Changes:**

- More ablation studies on the main methods (current stats/rejection sampling) should be added.
- The baseline comparison with real/synthesis data method (same quantity level) is encouraged, together with the compute cost comparison.
- Experimental settings on ImageNet (e.g., different loss w.r.t CIFAR; different image sizes) should be clarified.
- Hyper-parameters should be adjusted to give more discussions.

---

### Review · Reviewer_84PQ · 2026-07-05

**Summary Of Contributions:**

This paper studies noise-driven data-free knowledge distillation, where synthetic inputs are sampled from a standard Gaussian distribution and the student is trained to match the teacher’s outputs. In practice, this strategy works only when the teacher uses Batch Normalization (BN) in training mode (referred to as CS mode in the paper), but fails in evaluation mode (RS mode). The paper attributes this phenomenon to activation distribution shifts: the running statistics estimated from real images are mismatched with the batch statistics induced by random Gaussian inputs, leading to highly imbalanced class predictions. Motivated by this observation, the paper further proposes a batch-alignment wrapper to enable effective distillation from teachers using Layer Normalization (LN) or Group Normalization (GN).

**Audience:**

Yes

**Audience Explanation:**

The paper provides valuable insights into why noise-driven data-free knowledge distillation succeeds or fails under different normalization schemes and proposes a practical solution that broadens its applicability.

**Broader Impact Concerns:**

I do not see any ethical concerns or broader societal impacts that require additional discussion.

**Claims And Evidence:**

No

**Claims Explanation:**

1. **The central concept of “batch coupling” is not defined or analyzed rigorously enough.**
   Sections 3 and 4 provide an insightful empirical study of noise-driven distillation under different normalization strategies. However, the core concept of *batch coupling* is not developed with sufficient depth.
   - Could the authors provide a more rigorous definition of *batch coupling* and *inter-sample consistency*?
   - More importantly, could they elaborate on the underlying mechanism explaining why batch coupling is sufficient (or necessary) for successful noise-driven distillation? I think this part is not highlighted enough in the paper.
2. **The description of NormShift-KD is incomplete.**
   The first paragraph of Section 5 introduces NormShift-KD as consisting of a shared component, rejection sampling (RejS), BN-specific CS-mode inference, and a batch-alignment wrapper for LN/GN. However, several important components are insufficiently described.
   - Rejection sampling (RejS) is never introduced in the main text. Consequently, the meaning of the hyperparameter *oversample factor* k=4, as well as the distinction between **Gaussian+CS** and **NormShift-KD**, remains unclear.
   - Although CS-mode inference for BN is not a novel contribution of NormShift-KD, it should still be described in Section 5 for completeness, since it is presented as one of the method’s components.
3. **The normalization setup for the student network is unclear.**
   - Section 2.2 explicitly states that the student uses per-sample normalization. However, Sections 5.2 and 5.3 state that “BatchNorm-based students are used in all NormShift-KD experiments.” These statements appear inconsistent and should be clarified.
   - Since the student is distilled using random Gaussian inputs, what BatchNorm statistics are used during evaluation on real images? This point should also be explicitly explained.

**Requested Changes:**

Beyond the issues discussed above, I believe the paper would benefit from a clearer overall organization.

1. Figure 1 should include explicit legends explaining the RS/CS modes and the datasets used. The figure caption should also reference the corresponding experimental details (currently presented in Section 6.3). It may be more natural to move these experiments from Section 6.3 to Section 3.1, where the phenomenon is first detailedly introduced.
2. The analysis currently presented in Section 7.1 would fit better in Section 4 or Section 5, as the analysis of internal activations are directly related to the motivation and methodology.

Overall, I believe the paper would benefit from a clearer progression from empirical analysis, to method development, and finally to the main experimental evaluation.